# Beyond Chunking: Efficient Global Pooling for Holistic Long-Document Representation

## Abstract

Effectively representing long documents is a persistent challenge in natural language processing, as foundational encoders are constrained by limited context windows. Prevailing methods like chunking create fragmented representations that sever long-range dependencies and lose crucial global context, hindering downstream task performance. To overcome this, we introduce **Spectral Attention Token Pooling (SATPool)**, a novel, encoder-agnostic module that generates a single, holistic vector for a document of any length. SATPool operates in two stages: it first uses an efficient linear attention mechanism to capture global token interactions across the entire document, then employs a novel **Spectral Token Compression (STC)** technique to compress these globally-aware token representations into a compact, context-aware vector. We demonstrate that SATPool consistently and significantly outperforms established baselines through extensive experiments on diverse tasks, including long-document classification, Retrieval-Augmented Generation (RAG), multimodal RAG, and factuality consistency evaluation. Our work presents a practical, plug-and-play solution that unlocks the full potential of pre-trained encoders for long-form text without requiring costly retraining, enabling more robust document-level understanding and retrieval.

## 1 Introduction

Retrieval-Augmented Generation (RAG) (Lewis et al., 2020) has become a standard paradigm for enhancing the capabilities of Large Language Models (LLMs) (Li et al., 2024a) and, more recently, Large Multimodal Models (LMMs) by grounding them in external knowledge (Marino et al., 2019). These systems leverage vast repositories of information, from web pages to specialized legal (Wiratunga et al., 2024), medical (Raja et al., 2024), or technical documents (Mandanetwork et al., 2024), to generate more accurate and contextually relevant responses. However, a significant and persistent challenge arises when these source documents are longer than the input capacity of the text encoders used for retrieval. Foundational encoders like BERT (Devlin et al., 2019) and CLIP (Radford et al., 2021), which are widely used for their robust representations, are limited to relatively short contexts (e.g., 512 and 77 tokens, respectively), forcing a compromise when processing long-form text. Table 1 demonstrates that the long documents are prevalent in datasets for various tasks.

The predominant strategy to handle long documents is chunking, where a document is segmented into smaller, manageable pieces. A final representation is then created from these chunks. Some methods compute a single document vector by averaging the embeddings of all chunks. In RAG pipelines, other typical strategies include retrieving each chunk as an independent passage (Guu et al., 2020), or determining a document's relevance by the maximum similarity score across all of its chunks (Dai & Callan, 2019). While pragmatic, these strategies are fundamentally flawed. They create fragmented representations that sever long-range dependencies, thereby losing the overarching narrative and global context of the original document, as detailed in Section 3 and Figure 2.

To address this challenge, we introduce **Spectral Attention Token Pooling (SATPool)**, a novel pooling method that creates a single, holistic, fixed-size vector from a document of any length. It functions as an encoder-agnostic, plug-and-play module that enhances performance without requiring specialized long-context models or costly retraining. SATPool operates in two stages: first, it employs an efficient linear attention mechanism to capture global dependencies across the full sequence of tokens. Second, it uses a novel Spectral Token Compression (STC) technique to distill

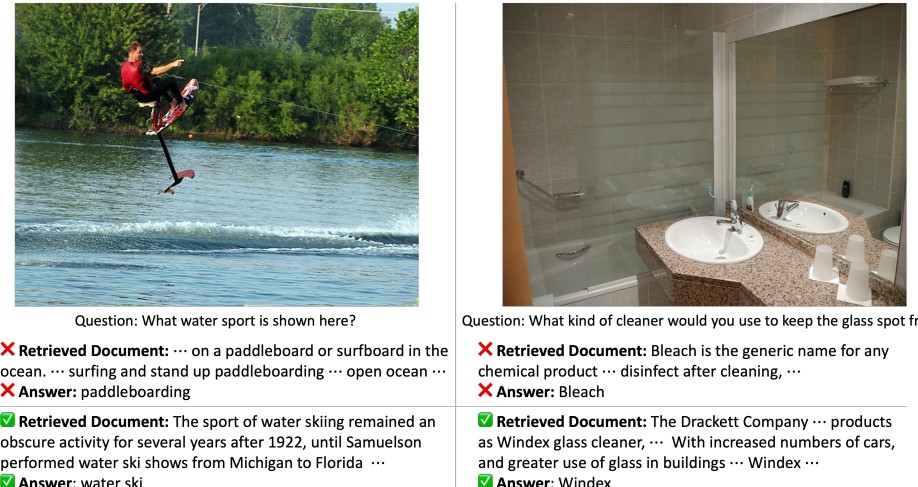

Figure 1: Qualitative comparison for Multimodal RAG. Our proposed pooling method, SATPool, consistently retrieves more relevant documents than mean pooling, leading to correct answer generation. In the left example, mean pooling retrieves a document about a generic water sport, whereas SATPool correctly identifies the passage about "water ski." Similarly, on the right, mean pooling finds a document about general cleaning solutions, while SATPool correctly retrieves the specific context for "glass cleaner." This demonstrates that SATPool captures nuanced, fine-grained context from the query, unlike mean pooling which often relies on superficial keyword matching.

these attention-reweighted embeddings into a compact representation that preserves both semantic content and high-level structural dynamics. In Figure 1, SATPool captures nuanced, fine-grained query context to retrieve semantically relevant documents, while mean pooling often relies on superficial keyword matching. We validate SATPool's effectiveness and ability to preserve holistic context through extensive experiments on a range of challenging tasks. Crucially, our work adopts a **representation-centric** perspective. While we validate our method on downstream tasks like RAG and text classification, our primary objective is not to optimize the generative components (e.g., the LLM in RAG), but to construct a **dense, holistic representation** for the document itself.

Our primary contributions are as follows:

1. We introduce **SATPool**, a novel, encoder-agnostic pooling method that generates a single, context-aware representation for documents of any length. Its two-stage architecture effectively captures both global semantics and structural dynamics.

2. We demonstrate through extensive experiments that SATPool achieves significant performance improvements over baselines across a range of diverse and challenging tasks, including long-document classification, RAG, multimodal RAG, and factuality consistency evaluation, proving its superior ability to maintain holistic contextual integrity.

Ultimately, SATPool provides a practical and effective solution to fully leverage pre-trained encoders on long-form text, improving performance and fidelity across a range of applications.

## 2 RELATED WORK

### 2.1 TEXT ENCODERS FOR LONG CONTEXT

The challenge of processing long sequences has spurred the development of specialized transformer architectures. Models like Longformer (Beltagy et al., 2020) and LongT5 (Guo et al., 2022) extended the context window to several thousand tokens (e.g., 4096 tokens) using efficient attention patterns. Benchmarks have consistently emphasized the need for such capabilities (Park et al., 2022). However, these models still operate under a fixed token limit. More recent work has produced models like

Table 1: Dataset analysis shows many documents exceed the text encoder's capacity (larger than 512 tokens), highlighting the need for global context pooling in long documents.

| Task | Dataset | # Instances | Avg. Length | Max. Length | % of Long Doc. |
|---|---|---|---|---|---|
| Text Classification | 20NewsGroups | 18,846 | 325.93 | 138,055 | 8.78 |
| | EURLEX | 60,000 | 1,509.01 | 248,252 | 60.18 |
| RAG | NQ | 39,415 | 3,766.93 | 68,842 | 89.56 |
| | HotPotQA | 88,287 | 2,202.71 | 49,503 | 72.22 |
| Multimodal RAG | OKVQA | 114,809 | 145.50 | 926 | 0.13 |
| | EVQA | 50,205 | 248.56 | 26,726 | 10.49 |
| Factuality Consistency | FactCC | 1,434 | 752.18 | 2,294 | 63.04 |
| | QAGS | 953 | 403.08 | 759 | 7.45 |

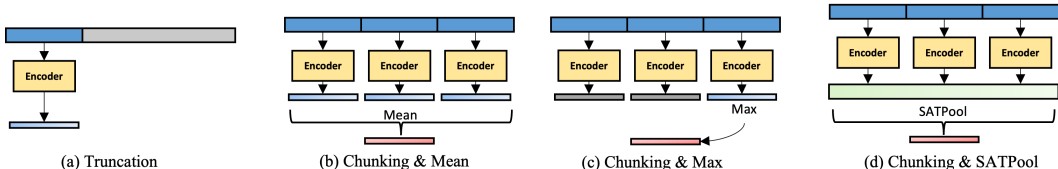

(a) Truncation     (b) Chunking & Mean     (c) Chunking & Max     (d) Chunking & SATPool

Figure 2: **(a)** Text encoders are constrained by limited input size, necessitating truncation for long documents. To accommodate longer inputs, chunking is typically employed. A common approach is **(b)** to average representations across chunks, or **(c)** to select the maximum scoring chunk, sometimes in RAG settings. However, such strategies inherently discard inter-chunk dependencies. In contrast, **(d)** SATPool captures global dependencies across the full token sequence without input length constraints, using linear attention followed by Spectral Token Compression (STC).

Longtriever (Yang et al., 2023), which is specifically pre-trained for long-context retrieval but is tied to its specific architecture, limiting its general applicability. Other approaches, such as LTR-BERT (Wang et al., 2024) and CFIR (Long et al., 2024), achieve improvement but require retraining the entire text encoder, which is computationally expensive and task-specific. A recent method, late chunking (Günther et al., 2024), also addresses this issue using context-aware chunking as a strategy to preserve locality, but still assumes the availability of an encoder capable of processing the entire document at once rather than as a necessity to overcome context limits. Our work is distinct in that it is encoder-agnostic and imposes no theoretical limit on document length.

## 2.2 HANDLING LONG CONTEXT IN RAG

Within the RAG research, handling long documents has been a persistent theme. The foundational RAG models for both NLP (Guu et al., 2020; Lewis et al., 2020; Liu et al., 2024) and multimodal tasks (Zhang et al., 2025) typically rely on the naive chunking strategy, where each chunk is treated as a separate passage for similarity search. This approach often fails to retrieve relevant information when the context required to understand a passage is located in a different chunk. More advanced methods have attempted to address this fragmentation; some create longer retrieval units by grouping documents but still rely on internal chunk-level scoring for relevance (Jiang et al., 2024), while others use post-retrieval modules to reconstruct context from an initial set of retrieved chunks (Zhao et al., 2024). In contrast, our method focuses on creating a single vector representation for the entire document, fundamentally changing how the document is indexed and compared.

While modern LLMs can handle longer input sequences, they aren't a complete solution to the problem. Research has shown that their performance drops significantly when relevant information is buried in the middle of a long context, known as the "lost in the middle" problem (Liu et al., 2023; Jin et al., 2024). This issue stems from a different aspect of long-context challenges, specifically related to generation tasks rather than creating representations.

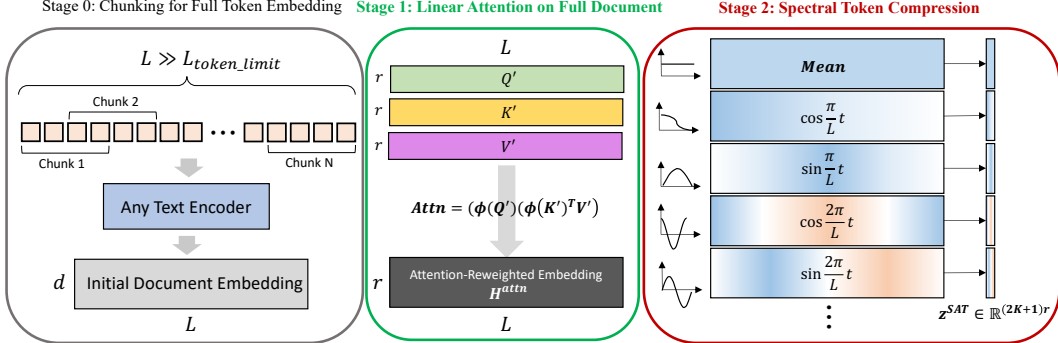

Figure 3: Overview of our proposed Spectral Attention Token Pooling (SATPool) pipeline using Performer attention and Spectral Token Compression (STC). **Stage 0**: The input document is first split into chunks to accommodate the token limit of the base encoder. Each chunk is encoded independently using a pretrained text encoder to form an initial document embedding of length $L$. **Stage 1:** Linear attention is applied to the full document embedding, yielding attention-reweighted token representations $\mathbf{H}^{\text{attn}}$ with efficient memory usage. **Stage 2**: STC performs a spectral pooling operation by computing the mean embedding and projecting the token sequence onto $K$-frequency sine and cosine bases. This yields a compact representation $\mathbf{z}^{\text{SAT}} \in \mathbb{R}^{(2K+1)r}$ that encodes both global semantics and coarse positional structure.

## 3 PROBLEM DEFINITION

Given a document $D$ represented as a long sequence of tokens with length $L$, and a pre-trained transformer-based encoder $\mathcal{E}$ with a maximum input capacity of $L_{\text{limit}}$ tokens, a fundamental challenge arises when $L \gg L_{\text{limit}}$.

Prevailing approaches resort to suboptimal strategies like truncation or chunking. With truncation, the document is simply cut off at the context limit, $D' = D_{1:L_{\text{limit}}}$, and its representation becomes $\mathbf{z}_{\text{trunc}} = \mathcal{E}(D')$. This approach leads to a catastrophic loss of any information beyond the initial segment. A more common strategy is chunking, where the document $D$ is partitioned into a set of $N$ smaller chunks, $D = \{\mathbf{c}_1, \mathbf{c}_2, \ldots, \mathbf{c}_N\}$, such that $|\mathbf{c}_i| \leq L_{\text{limit}}$. The encoder processes each chunk independently, and the resulting representations are aggregated. For instance, in text classification, a document vector is often computed by averaging chunk representations, $\mathbf{z}_{\text{mean}} = \frac{1}{N} \sum_{i=1}^{N} \mathcal{E}(\mathbf{c}_i)$. Similarly, in RAG and Multimodal RAG, a document's relevance to a query $\mathbf{q}$ is typically determined by the maximum similarity score over its constituent chunks, $\text{score}(D, \mathbf{q}) = \max_{i=1,\ldots,N} \text{sim}(\mathcal{E}(\mathbf{c}_i), \mathcal{E}(\mathbf{q}))$. However, both truncation and chunking-based aggregation are fundamentally flawed. Truncation discards vast portions of the text, while chunking severs long-range dependencies and discards the overarching narrative, resulting in a fragmented representation of the document's global context, as illustrated in Figure 2.

Our objective is to design a pooling function, $f_{pool}$, that acts as a lightweight, plug-and-play module operating on top of a frozen encoder. This function maps the full sequence of token embeddings $H \in \mathbb{R}^{L \times d}$ to a single, fixed-size vector $z \in \mathbb{R}^d$. We define $z$ as a **dense, holistic representation** of the document. Unlike standard pooling which aggregates local signals, $z$ must capture the document's *semantic trajectory* and structural dynamics.

## 4 PROPOSED METHOD

To address the challenges of processing long documents, we propose **Spectral Attention Token Pooling (SATPool)**, a novel two-stage method that generates a fixed-size, context-aware representation from a sequence of any length. Our approach operates on the hidden states produced by a standard, frozen pre-trained encoder, making it a plug-and-play enhancement for existing pipelines.

## 4.1 STAGE 1: LINEAR ATTENTION FOR GLOBAL CONTEXT

Given a sequence of hidden states $\mathbf{H} \in \mathbb{R}^{L \times d}$ obtained from the encoder by chunking with sliding window (Stage 0), we first capture global interactions across all $L$ tokens. To achieve this efficiently, we employ a linear attention mechanism suggested by Performer (Choromanski et al., 2020).

Let's recall Attention mechanism (Vaswani et al., 2017) is defined as

$$\text{Attn}(\mathbf{Q}, \mathbf{K}, \mathbf{V}) = \text{softmax}\left(\mathbf{Q}\mathbf{K}^T/\sqrt{d}\right)\mathbf{V} = \mathbf{D}^{-1}\mathbf{A}\mathbf{V}, \ \mathbf{A} = \exp(\mathbf{Q}\mathbf{K}^T/\sqrt{d}), \ \mathbf{D} = \text{diag}(\mathbf{A}\mathbf{1}_L),$$

requiring the $\mathcal{O}(L^2 d)$ time and $\mathcal{O}(L^2)$ memory complexity for $\mathbf{Q}, \mathbf{K}, \mathbf{V} \in \mathbb{R}^{L \times d}$. The Performer approximates the attention matrix $\mathbf{A}$ by decomposing the softmax kernel using a mapping function $\phi : \mathbb{R}^d \to \mathbb{R}^r$ such that $\exp\left(\frac{\mathbf{q}_i^T \mathbf{k}_j}{\sqrt{d}}\right) \approx \phi(\mathbf{q}_i)^T \phi(\mathbf{k}_j)$. This allows the attention to be computed with a different order of operations, reducing the complexity to linear:

$$\text{Attn}(\mathbf{Q}, \mathbf{K}, \mathbf{V}) \approx \hat{\mathbf{D}}^{-1}\phi(\mathbf{Q})[\phi(\mathbf{K})^T \mathbf{V}], \ \hat{\mathbf{D}} = \text{diag}(\phi(\mathbf{Q})\phi(\mathbf{K})^T \mathbf{1}_L), \tag{1}$$

resulting in $\mathcal{O}(Ldr)$ and $\mathcal{O}(Lr)$ time and memory complexity. The detailed derivation for Performer is introduced in Appendix A.1.

To this end, we utilize Performer to recompute the full attention on hidden states $\mathbf{H} \in \mathbb{R}^{L \times d}$. Query, key, and value matrices are projected to low-rank, $\mathbf{Q}' = \mathbf{H}\mathbf{W}_q, \mathbf{K}' = \mathbf{H}\mathbf{W}_k, \mathbf{V}' = \mathbf{H}\mathbf{W}_v \in \mathbb{R}^{L \times r}$ (where $r \ll d$), using trainable projection matrices $\mathbf{W}_{\{q,k,v\}} \in \mathbb{R}^{d \times r}$. In practice, the positive-valued mapping function $\phi(\mathbf{x}) = \text{ReLU}(\mathbf{x}) + \epsilon$ is selected in (Choromanski et al., 2020). The final global attention-reweighted sequence is obtained by

$$\mathbf{H}^{\text{attn}} = \hat{\mathbf{D}}^{-1}\phi(\mathbf{Q}')[\phi(\mathbf{K}')^T \mathbf{V}'] \in \mathbb{R}^{L \times r}. \tag{2}$$

Therefore, this linear complexity overcomes the quadratic bottleneck of standard attention, making the process scalable to very long sequences while effectively modeling token inter-dependencies across the entire document. Figure 4 demonstrates the comparison between chunking strategy and full linear attention.

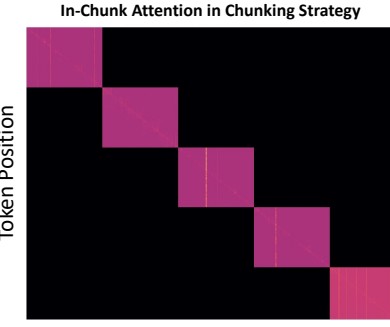

Figure 4: A comparison of attention mechanisms on a long document. (Left) The standard chunking strategy results in a fragmented, block-diagonal attention map, where connections are confined within each chunk. (Right) Our proposed linear attention operates over the full sequence of token embeddings, creating a complete, global attention map that captures long-range dependencies across the entire document.

This stage effectively bridges the gap caused by chunking. While the initial encoding in Stage 0 is local, the semantic information is preserved within the high-dimensional embeddings. The linear attention mechanism serves as a global relational discovery process: it scans the entire sequence to identify semantic correlations between distant parts (e.g., connecting a subject in the first chunk to a reference in the fifth chunk) before any information is compressed. This allows SATPool to recover global dependencies that are effectively "distributed" across the independent chunks.

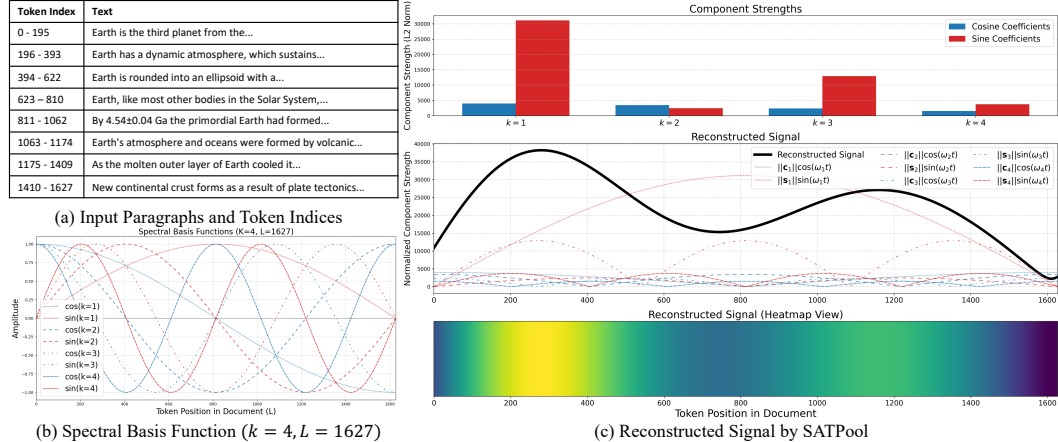

(a) Input Paragraphs and Token Indices

(b) Spectral Basis Function ($k = 4, L = 1627$)

(c) Reconstructed Signal by SATPool

Figure 5: **An illustration of the Spectral Token Compression (STC) process.** **(a)** The input document is represented by its paragraphs. **(b)** STC uses a fixed set of unweighted sine and cosine **basis functions**. **(c)** Through training, the model learns the **strengths** of these components (top bar chart). The summation of these weighted components to create a **reconstructed signal** (middle and bottom) which highlights the regions of the document the model deems most structurally important.

## 4.2 STAGE 2: SPECTRAL TOKEN COMPRESSION (STC)

To transform the variable-length token sequence $\mathbf{H}^{\text{attn}} \in \mathbb{R}^{L \times r}$ into a compact, fixed-size vector representation, we propose *Spectral Token Compression (STC)*, a pooling strategy that captures both semantic content and structural dynamics of the document. Unlike mean-pooling or **[CLS]**-token aggregation, which treat the sequence as an unordered set of embeddings, STC preserves position-sensitive information by interpreting the sequence as a multi-dimensional discrete signal.

Specifically, each token embedding $\mathbf{H}_t^{\text{attn}} \in \mathbb{R}^r$ at position $t = 1, \dots, L$ is treated as a point sampled from a $r$-dimensional signal. STC first computes the global semantic content by taking the mean, $\bar{\mathbf{H}}^{\text{attn}} = \frac{1}{L} \sum_{t=1}^{L} \mathbf{H}_t^{\text{attn}}$. Then, to capture positional structure, it projects the signal onto a set of $K$ sine and cosine bases at frequencies $\omega_k = \frac{\pi k}{L}$.

$$\mathbf{C}_k = \sum_{t=1}^{L} \mathbf{H}_t^{\text{attn}} \cos(\omega_k t), \quad \mathbf{S}_k = \sum_{t=1}^{L} \mathbf{H}_t^{\text{attn}} \sin(\omega_k t), \quad k = 1, \dots, K. \tag{3}$$

The cosine and sine coefficient vectors $\{\mathbf{C}_k, \mathbf{S}_k\}$ act as learned, high-dimensional "importance weights" for each of the fixed basis functions. Through end-to-end training, the model learns to assign larger magnitudes to the coefficients that correspond to the most structurally significant parts of the document. For example, a low-frequency component like 'cos(k=1)' inherently assigns the most weight to the beginning of the sequence, making it ideal for capturing the structural importance of an introduction. Similarly, 'sin(k=1)' peaks in the middle of the sequence, allowing the model to emphasize the main body of the document. Crucially, by learning a weighted combination of these and higher-frequency components, the model can create a **reconstructed signal** that highlights any region or combination of regions. As shown in Figure 5(c) and analyzed in detail in Section 4.3, the peaks in this final signal highlight the regions of the document that the model has identified as having the highest **structural importance**. Finally, the mean vector ($\bar{\mathbf{H}}^{\text{attn}}$) and the $2K$ spectral coefficients (Eq.3) are concatenated to form the final representation,

$$\mathbf{z}^{\text{SAT}} = \left[ \bar{\mathbf{H}}^{\text{attn}}, \mathbf{C}_1, \mathbf{S}_1, \dots, \mathbf{C}_K, \mathbf{S}_K \right] \in \mathbb{R}^{(2K+1)r}. \tag{4}$$

This fixed-size vector $\mathbf{z}^{\text{SAT}}$ serves as a rich and compact representation of the long document input. Concatenation preserves the distinct information captured by each component—the global semantics from the mean and the various structural patterns from each spectral frequency. A subsequent projection layer, $\mathbf{W}_o \in \mathbb{R}^{(2K+1)r \times d}$, then learns the optimal combination of these rich features to produce the final document embedding, $\mathbf{z}^{\text{SAT}}\mathbf{W}_o = \mathbf{z} \in \mathbb{R}^d$. Importantly, the dimensionality of this vector is independent of the original sequence length $L$, making it well-suited for downstream classification or retrieval tasks that require fixed-length input.

Table 2: Experimental results (F1 score) for text classification task for 20News and EURLEX datasets.

| Methods | 20NewsGroups | | | | EURLEX | | | |
|---|---|---|---|---|---|---|---|---|
| | BERT | RoBERTa | Longformer | LongT5 | BERT | RoBERTa | Longformer | LongT5 |
| CLS | 0.5916 | 0.5890 | 0.5798 | 0.2606 | 0.5645 | 0.4975 | 0.4933 | 0.4411 |
| Mean | 0.6223 | 0.6315 | 0.6386 | 0.5637 | 0.6856 | 0.6515 | 0.6644 | 0.6921 |
| LC | 0.6166 | 0.6115 | 0.6219 | 0.4909 | 0.6763 | 0.6708 | 0.6731 | 0.4960 |
| PARADE | 0.5819 | 0.6070 | 0.6258 | 0.2318 | 0.6318 | 0.6484 | 0.6609 | 0.3857 |
| TrLDC | 0.5754 | 0.5705 | 0.5733 | 0.2203 | 0.6277 | 0.6341 | 0.6431 | 0.4969 |
| **SATPool** | **0.6371** | **0.6570** | **0.6450** | **0.6195** | **0.7124** | **0.7266** | **0.7232** | **0.7306** |

Table 3: Experimental results for RAG for NQ and HotPotQA datasets. **Bold** indicates the best result for each baseline, while underline denotes the second best.

| Model / Method | | NQ | | | | HPQA | | | |
|---|---|---|---|---|---|---|---|---|---|
| | | R@1 | R@5 | F1 | EM | R@1 | R@5 | F1 | EM |
| BERT-DPR | Baseline | 0.0670 | 0.1935 | 0.1549 | 0.1030 | 0.0150 | 0.0490 | 0.1932 | 0.1250 |
| | Mean | 0.2000 | 0.4281 | 0.1846 | 0.1290 | 0.0940 | 0.1960 | 0.2124 | 0.1320 |
| | MaxP | 0.0605 | 0.2778 | 0.1556 | 0.1030 | 0.0290 | 0.1220 | 0.2242 | 0.1460 |
| | LC | 0.1838 | 0.4270 | 0.1976 | 0.1391 | 0.0980 | 0.2120 | **0.2401** | **0.1720** |
| | **SATPool** | **0.2962** | **0.5589** | **0.2361** | **0.1850** | **0.1060** | **0.2200** | 0.2262 | 0.1590 |
| GTR-T5 | Baseline | 0.5049 | 0.7762 | 0.2878 | 0.2212 | 0.4070 | 0.5890 | **0.3120** | **0.2190** |
| | Mean | 0.4973 | 0.7578 | 0.2862 | 0.2252 | 0.3220 | 0.5310 | 0.2812 | 0.1900 |
| | MaxP | 0.1449 | 0.5730 | 0.2484 | 0.1860 | 0.0600 | 0.3100 | 0.2732 | 0.1870 |
| | LC | 0.3341 | 0.6162 | 0.2538 | 0.1880 | 0.2460 | 0.4100 | 0.2667 | 0.1830 |
| | **SATPool** | **0.5395** | **0.7914** | **0.3022** | **0.2282** | **0.4180** | **0.5980** | 0.2992 | 0.2020 |

**Comparison with Classical Spectral Analysis.** While STC draws inspiration from classical signal processing (e.g., Discrete Fourier Transform), it differs fundamentally in its adaptability. Classical spectral filters are static and fixed. In contrast, STC is a fully differentiable pooling layer integrated into the network's computation graph. While the basis functions (sine and cosine) provide a fixed "ruler," the coefficient vectors are learned dynamically via backpropagation. This allows the model to identify and weight structural components (e.g., introductions or conclusions) based on their semantic importance to the specific downstream task, rather than their absolute frequency alone.

## 4.3 ANALYZING STC WITH VISUALIZATION

To provide an intuitive understanding of how STC operates, we offer a visualization of the process in Figure 5. The figure begins with a sample document about 'Earth', where the first two paragraphs serve as the introduction **(a)**. STC projects the document's token embeddings onto a set of fixed sine and cosine basis functions **(b)**, corresponding to the components in Eq 3. While our model concatenates the learned coefficient vectors ($\mathbf{C}_k, \mathbf{S}_k \in \mathbb{R}^r$) to form the final representation, we can visualize their importance to understand the model's behavior. We measure the "strength" of each coefficient vector by taking its L2 norm, with the results shown in the bar chart in **(c)**. The bar chart reveals the learned importance of each basis function, but the final reconstructed signal is a result of their combined constructive and destructive interference. This final signal highlights the regions the model deems most structurally important—in this case, the introduction. For illustrative purposes, we create a reconstructed signal by summing the basis functions weighted by these strengths (e.g., $||\mathbf{C}_k|| \cos(\omega_k t)$). The middle plot displays both the individual weighted components and their final summation as a solid black line. The bottom plot shows a heatmap view of this same reconstructed signal for visual clarity. This demonstrates how STC learns to identify and weigh key sections of a document by combining multiple spectral components.

Table 4: Experimental results for MultiModal-RAG for OKVQA and EVQA datasets. **Bold** indicates the best result for each baseline.

| Model / Method | | OKVQA | | | | EVQA | | | |
| --- | --- | --- | --- | --- | --- | --- | --- | --- | --- |
| | | PR@5 | PR@10 | VQAScore | EM | R@5 | R@10 | VQAScore | EM |
| CLIP-DPR | Baseline | 0.1344 | 0.1912 | 0.1980 | 0.2216 | 0.3114 | 0.4103 | 0.0785 | 0.2354 |
| | Mean | 0.1738 | 0.2457 | 0.2074 | 0.2299 | 0.3023 | 0.4091 | 0.0756 | 0.2269 |
| | MaxP | 0.0862 | 0.1336 | **0.2509** | **0.2788** | 0.0337 | 0.1634 | 0.0760 | 0.2280 |
| | LC | 0.1768 | 0.2539 | 0.2083 | 0.2325 | 0.2337 | 0.3120 | 0.0703 | 0.2109 |
| | **SATPool** | **0.2008** | **0.2808** | 0.2208 | 0.2446 | **0.3360** | **0.4286** | **0.0808** | **0.2423** |
| RAVQA | Baseline | 0.0541 | 0.0906 | 0.2806 | 0.3084 | 0.0909 | 0.1377 | 0.0524 | 0.1571 |
| | Mean | 0.1314 | 0.1930 | 0.2914 | 0.3189 | 0.1394 | 0.1960 | **0.0575** | **0.1726** |
| | MaxP | 0.1092 | 0.1798 | **0.2960** | **0.3238** | 0.0177 | 0.0794 | 0.0564 | 0.1691 |
| | LC | 0.1344 | 0.1964 | 0.2878 | 0.3161 | 0.1389 | 0.1920 | **0.0575** | **0.1726** |
| | **SATPool** | **0.1582** | **0.2279** | 0.2859 | 0.3145 | **0.1549** | **0.2057** | 0.0573 | 0.1720 |

## 5 EXPERIMENTAL DETAILS

We validate our approach on four diverse downstream tasks that highlight the challenges of long-document processing: text classification, Retrieval-Augmented Generation (RAG), Multimodal RAG, and Factuality Consistency Evaluation . These tasks frequently contain documents exceeding standard encoder input limits, as shown in Table 1. To ensure a fair comparison, our experiments use frozen pre-trained encoders, allowing us to directly measure how different pooling and chunking strategies affect the quality of the final document representation. Detailed descriptions of the task setups, datasets, and evaluation metrics are available in Appendix C and D, while the factuality consistency evaluation is introduced in Appendix E separately.

### 5.1 COMPARISONS

We evaluate SATPool against two main categories of methods for handling long documents. First, we compare against specialized long-context models that are architecturally designed to process long sequences. For text classification, this includes models like Longformer (Beltagy et al., 2020) and LongT5 (Guo et al., 2022).

Second, we compare against model-agnostic approaches that, like SATPool, are designed as plug-and-play modules for standard encoders. Our primary comparison in this category is Late Chunking (Günther et al., 2024), a versatile method that we evaluate across text classification, RAG, and multi-modal RAG tasks. We also compare against hierarchical aggregation methods, specifically PARADE (Li et al., 2023) and TrLDC (Dai et al., 2022), which utilize transformer-based layers to aggregate chunk information. Additionally, we benchmark against standard baseline strategies, including simple mean pooling, MaxP (Dai & Callan, 2019), and using the `[CLS]` token representation from the encoder. The details of these comparison methods are introduced in Appendix G.

### 5.2 STATISTICAL SIGNIFICANCE AND ROBUSTNESS

To ensure the stability of our results and strictly validate the performance gains, we adopt simple fold-based evaluation in favor of bootstrapping. For all RAG experiments reported in Table 11, we evaluate on the full combined Validation and Test sets (e.g., 4,280 samples for NQ). The reported performance metrics (Mean $\pm$ Standard Deviation) are derived from 1,000 bootstrap resamples of this full evaluation set. This rigorous protocol confirms that SATPool's improvements are statistically robust across the data distribution and not artifacts of specific data subsets.

## 6 RESULT ANALYSIS

### 6.1 EXPERIMENTAL RESULTS

As detailed in Tables 2, 3, and 4, SATPool demonstrates consistent and significant performance gains across every tested task, dataset, and model. It consistently outperforms standard pooling methods

like mean pooling, as well as more advanced techniques such as Late Chunking. This advantage is evident in the enhanced F1 scores for text classification and the boosted retrieval metrics (Recall and Pseudo-Relevance Recall, Appendix D) in both RAG and Multimodal RAG settings. The details about factuality consistency evaluation task and its results are introduced in Appendix E.

The qualitative results in Figure 1 further illustrate SATPool's advantage in retrieval. The examples show that SATPool captures nuanced, fine-grained query context to retrieve semantically relevant documents, whereas mean pooling often relies on superficial keyword matching.

**Limitation.** However, while SATPool substantially improves document retrieval, these gains do not always lead to improved generation performance, such as the Exact Match (EM) score in some RAG tasks. This observation is consistent with the well-documented "needle-in-a-haystack" problem (Laban et al., 2024; Jin et al., 2024), where generator models struggle to pinpoint the correct information within a long context, even when it is successfully retrieved.

## 6.2 ABLATION STUDY

### 6.2.1 CONTRIBUTIONS OF EACH MODULE

To validate the contribution of each module in SATPool, e.g., linear attention, STC, and the impact of $K$, we conduct ablation studies adjusting the condition on the text classification task. We report the experimental result on 20News and EURLEX dataset with BERT and RoBERTa by comparing the baseline (`[CLS]`), mean pooling, STC only varying $k$, Attention-only, and SATPool varying $k$. The ablation study results in Figure 6 show that pooling is much better than relying on `[CLS]` tokens, and adding linear attention particularly improves the classification performance. The effectiveness of STC depends on the characteristic of dataset. For example, STC has a negligible impact on the 20News dataset but a significant one on the EURLEX dataset. This may be because the documents in 20News are relatively short and have a simpler content structure compared to those in the EURLEX.

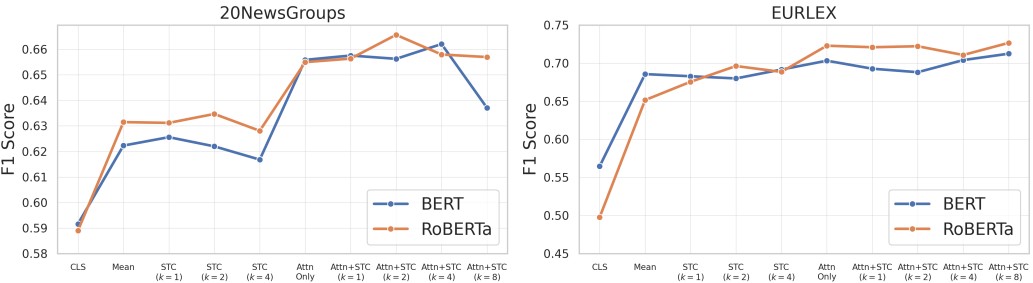

Figure 6: Ablation studies for 20NewsGroups and EURLEX datasets with frozen BERT and RoBERTa backbone networks. Compared to the baseline relying on `[CLS]` tokens, plugging in full-attention layer and STC significantly improve the classification performance.

### 6.2.2 ARCHITECTURAL CONTRIBUTION VS. PARAMETER COUNT

SATPool's architecture introduces additional trainable parameters. This raises a critical question: do the performance gains stem from our novel design, or are they merely a byproduct of an increased parameter count? To isolate the architectural benefits, we compare SATPool against a baseline mean-pooling model equipped with a significantly larger projection layer, denoted as 'Mean-Large'. As shown in Table 5, this baseline uses a larger hidden dimension (1024 vs. 256) in the three-layer MLP classifier to ensure its parameter count is comparable to, or greater than, our SATPool models.

The results clearly show that SATPool consistently outperforms this parameter-matched baseline. For example, 'SATPool ($k = 8$)' achieves an F1 score of 0.7266 on the EURLEX dataset with RoBERTa, surpassing the 'Mean-Large' (0.7039) while using fewer parameters (1.51M vs. 1.59M). This demonstrates that the improvements are attributable to SATPool's novel architecture for capturing global context, rather than simply an increase in the number of parameters. Furthermore, these

additional parameters are negligible relative to the size of the encoders, adding less than 1.5% to the total parameter count of BERT, highlighting the efficiency of our approach.

Table 5: Comparison of SATPool against mean-pooling baselines with varying parameter counts on the 20News and EURLEX datasets. The 'Mean-Large' model uses an enlarged projection layer to provide a parameter-matched baseline. Best F1 scores for each backbone and dataset are in **bold**.

| Method | Hidden Dim. | # of Param. | Added Params. (% of Backbone, BERT/RoBERTa) | 20News | | EURLEX | |
|---|---|---|---|---|---|---|---|
| | | | | BERT | RoBERTa | BERT | RoBERTa |
| Mean | 256 | 0.20M | 0.18%/0.06% | 0.6223 | 0.6315 | 0.6856 | 0.6515 |
| Mean-Large | 1024 | **1.59M** | **1.45%/0.45%** | 0.6243 | 0.6289 | 0.6977 | 0.7039 |
| SATPool ($k=4$) | 256 | 0.99M | 0.90%/0.28% | **0.6621** | **0.6580** | 0.7040 | 0.7107 |
| SATPool ($k=8$) | 256 | 1.51M | 1.38%/0.43% | 0.6371 | 0.6570 | **0.7124** | **0.7266** |

While impact of $k$ in SATPool is illustrated in Figure 6 and explained in Section 6.2.1, the impact of the low rank size $r$ is introduced in Appendix F.2.

## 6.3 COMPUTATIONAL EFFICIENCY ANALYSIS

To demonstrate the practical viability of SATPool, we conducted rigorous wall-clock and FLOPs benchmarking on a realistic long-document scenario (19,502 tokens). We compared SATPool against standard baselines, hierarchical transformers, and specialized long-context models using a single NVIDIA RTX A5000 GPU.

**Runtime and Scalability.** As detailed in Table 14 (Appendix H.1), SATPool incurs negligible runtime overhead compared to simple Mean Pooling (1347ms vs. 1333ms). This indicates that the computational cost is dominated by the backbone encoder processing chunks, rendering our $O(L)$ linear attention and STC operations effectively free in real-world pipelines. Notably, SATPool is approximately 18% faster than Late Chunking (1657ms) and 1.7x faster than the specialized Longformer architecture (2356ms). This validates our core motivation: adapting efficient short-context encoders via SATPool is more scalable than deploying heavy, specialized long-context architectures.

**Memory Footprint.** Our memory analysis (Appendix H.2) confirms that the bottleneck in long-context RAG is the generator, not the pooling mechanism. SATPool increases total system memory usage by only 1.27% relative to the Truncation baseline, fitting easily within standard hardware constraints.

## 7 CONCLUSION

In this work, we address the persistent challenge of creating holistic representations for long documents, a common scenario where standard methods like truncation and chunking result in significant information loss and fragmented context. We introduce Spectral Attention Token Pooling (SATPool), a novel, two-stage module that serves as a lightweight, plug-and-play solution for any pre-trained text encoder. By first applying an efficient linear attention mechanism to capture global dependencies and then using Spectral Token Compression (STC) to distill the sequence into a fixed-size vector, SAT-Pool effectively preserves structural integrity of the original document. Our extensive experiments demonstrate that SATPool consistently and significantly outperforms established baselines across a diverse range of applications, including long-document classification, RAG, multimodal RAG, and factuality evaluation. Ultimately, SATPool provides a practical and powerful method for unlocking the full potential of pre-trained encoders on long-form text. By producing a single, globally-aware vector for documents of any length, it enables more effective indexing, retrieval, and analysis, paving the way for more capable and context-aware NLP systems.

## ETHICS STATEMENT

We have adhered to the Code of Ethics for academic publishing in this work. All experiments were conducted on publicly available and widely used academic benchmark datasets, as detailed in Appendix C. No new data involving human subjects was collected. Our proposed method, SATPool, is a general-purpose module designed for standard NLP and multimodal tasks on established academic benchmarks. Within the scope of this work, we have not identified any direct negative societal impacts or ethical concerns arising from our methodology or its applications.

## REPRODUCIBILITY STATEMENT

We are committed to ensuring the reproducibility of our research. The architecture and methodology of our proposed SATPool module are described in detail in Section 4, including the linear attention mechanism (Section 4.1) and Spectral Token Compression (Section 4.2). All experimental setups, including dataset descriptions, sampling procedures, and hyperparameters such as learning rates, batch sizes, and model dimensions for all four tasks are provided in Appendix C. The evaluation metrics used for our experiments are formally defined in Appendix D. To further facilitate verification and future work, we have included our source code in the supplementary material.

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

# A  A Deeper Look into Performer and Linear Attention

## A.1  Performer Attention (Choromanski et al., 2020)

For query, key, and value matrices $\mathbf{Q}, \mathbf{K}, \mathbf{V} \in \mathbb{R}^{L \times d}$, the original attention mechanism computes

$$\text{softmax}\left(\frac{\mathbf{Q}\mathbf{K}^{\top}}{\sqrt{d}}\right)\mathbf{V}.$$

Equivalently, it can be written as

$$\text{Attn}(\mathbf{Q}, \mathbf{K}, \mathbf{V}) = \mathbf{D}^{-1}\mathbf{A}\mathbf{V}, \quad \mathbf{A} = \exp\left(\frac{\mathbf{Q}\mathbf{K}^{\top}}{\sqrt{d}}\right), \quad \mathbf{D} = \text{diag}(\mathbf{A}\mathbf{1}_L).$$

This computation requires $\mathcal{O}(L^2 d)$ time and $\mathcal{O}(L^2)$ memory, which is prohibitive for long sequences. To reduce this cost, Performer approximates the softmax kernel $SM(x, y) = \exp(x^{\top}y)$ using a kernel of the form

$$K(x, y) = \mathbb{E}_{\omega}[\phi(x)^{\top}\phi(y)],$$

where $\phi : \mathbb{R}^d \to \mathbb{R}^r_+$ is a random feature map. This approximation is unbiased, i.e., the expectation of the dot product of the random features equals the softmax kernel. Choromanski et al. (2020) proposes a provable positive random feature map:

$$\phi(x) = \exp(\omega^{\top}x - \|x\|^2/2), \quad \omega \sim \mathcal{N}(0, I),$$

which satisfies the unbiasedness condition for approximating the softmax kernel.

In addition, Choromanski et al. (2020) explored generalized attention mechanisms, where other positive feature maps can be used for efficiency. For instance, they show that a simple choice like

$$\phi(x) = \text{ReLU}(x) + \epsilon$$

can be used as a kernel approximation, even though it does not approximate softmax theoretically. This choice is empirically effective, particularly for long sequence tasks like protein modeling.

Finally, using the decomposition $\mathbf{Q}' = \phi(\mathbf{Q})$, $\mathbf{K}' = \phi(\mathbf{K})$, the approximate attention becomes

$$\text{Attn}(\mathbf{Q}, \mathbf{K}, \mathbf{V}) \approx \hat{\mathbf{D}}^{-1}\mathbf{Q}'(\mathbf{K}'^{\top}\mathbf{V}), \quad \hat{\mathbf{D}} = \text{diag}(\mathbf{Q}'(\mathbf{K}'^{\top}\mathbf{1}_L)).$$

This formulation reduces the time complexity to $\mathcal{O}(Lrd)$ and the space complexity to $\mathcal{O}(Lr + Ld + rd)$, where $r$ is the number of random features.

# B  Robustness of Spectral Token Compression

To validate the robustness of STC in identifying important regions within a document, we intentionally altered the order of the document introduced in Figure 5. Specifically, we reversed the order of its paragraphs, moving the original introduction to the second half of the text. Since STC previously highlighted the introduction at the beginning of the document, a robust model should now identify this same content in its new position.

As shown in Figure 7, the reconstructed signal confirms this hypothesis. STC still identifies the content of the original introduction as the most structurally important region, despite its new location in the latter half of the document. This result demonstrates that STC is sensitive to the document's semantic content rather than relying on fixed positional patterns.

# C  Implementation Details

**Long Text Classification.** For multi-class classification, we use the 20NewsGroups (Lang, 1995) dataset, which consists of 11.3k training and 7.5k test instances. Moreover, for multi-label classification, we use the EURLEX (Chalkidis et al., 2019) dataset of EU legal documents, as used in (Park et al., 2022). For the EURLEX dataset, we utilize the English version, which contains 55k training and 5k test instances; for computational efficiency, we sample 10k instances for training in all experiments.

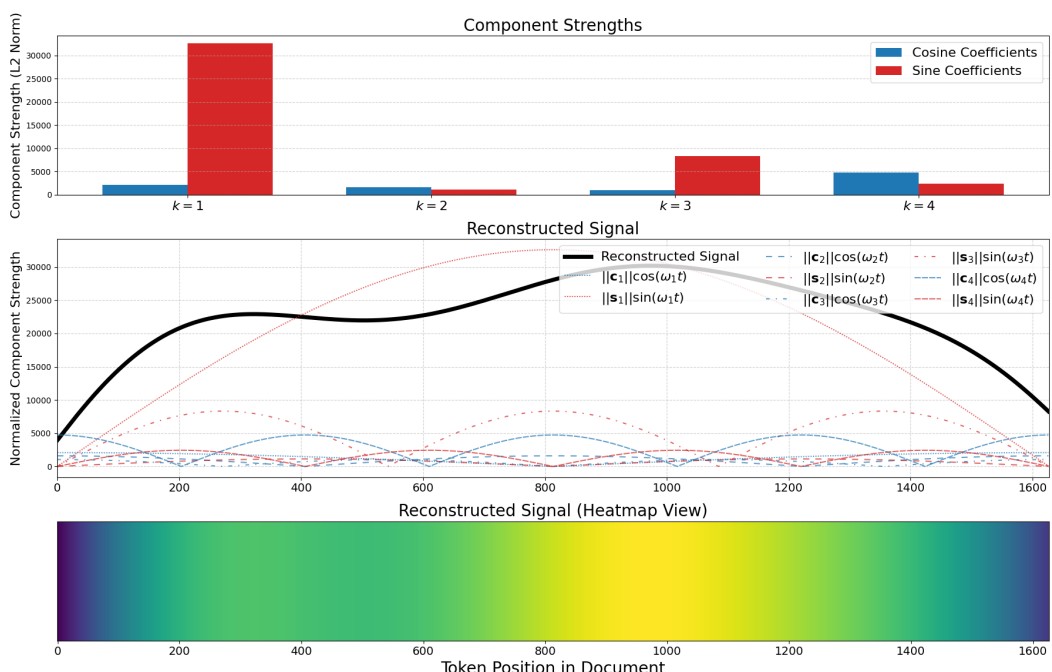

Figure 7: **Robustness analysis of Spectral Token Compression (STC)**. The paragraph order of the document from Figure 5 was reversed, moving the original introduction to the second half of the text. The reconstructed signal shows that STC still identifies the content of the original introduction as the most structurally important region, despite its new position. This demonstrates that STC is sensitive to the document's semantic content rather than fixed positional patterns.

Our classification setup uses frozen, pre-trained encoders (BERT, RoBERTa, Longformer, and LongT5) followed by a three-layer MLP classifier with a hidden dimension of 256. All models are trained for 50 epochs using an Adam optimizer with a learning rate of $10^{-4}$ and a batch size of 16. Crossentropy loss is used for multi-class (20NewsGroups) classification, and Binary cross entropy loss is used for multi-label classification.

**RAG with Long Documents.** We evaluate our RAG pipeline on the Natural Questions (NQ) (Kwiatkowski et al., 2019) and HotPotQA (Yang et al., 2018) datasets, as used in (Li et al., 2024b; Petroni et al., 2020). Both use Wikipedia as their knowledge source. For efficiency, we use a subset of 20k training samples and 1k test samples for both datasets. Our training process consists of two stages: retriever training followed by generator fine-tuning.

For the retriever, we build a DPR-style model using frozen BERT and GTR-T5 encoders. Separate three-layer MLP projection heads with an output dimension of 512 are trained for queries and documents. The retriever is trained for 100 epochs with a batch size of 16, using an Adam optimizer and a learning rate of $10^{-5}$. The retriever is trained using a Supervised Contrastive (SupCon) (Khosla et al., 2020) loss function.

For the generator, we fine-tune a T5-large (Raffel et al., 2020) model for 10 epochs with a batch size of 2 and a learning rate of $5 \times 10^{-5}$, optimizing a cross-entropy loss objective. Since the full text of retrieved documents exceeds the generator's input capacity, the top 5 retrieved documents are first summarized using a pre-trained BART (Lewis et al., 2019) model. These summaries are then concatenated and provided as context to the generator.

**Multimodal RAG with Long Documents.** For multimodal RAG, we use the OKVQA (Marino et al., 2019) and EVQA (Mensink et al., 2023) datasets, following the setup suggested in (Lin et al., 2024). The OKVQA dataset pairs 9.0k training and 5.1k test samples with 115k documents, while EVQA contains 167k training and 3.8k test samples paired with 50.2k documents.

We evaluate two retriever architectures. The first is a simple CLIP-based DPR model that we created. The second is RAVQA Lin & Byrne (2022), which uses a more complex query encoder that incorporates image representations from CLIP, OCR, and image descriptions. For both models, we use a frozen text encoder and train a three-layer MLP projection head with a hidden dimension of 512 for both queries and documents. The CLIP-DPR model was trained with a learning rate of $10^{-4}$, while RAVQA used $10^{-5}$. Both were trained for 100 epochs with a batch size of 16 using an Adam optimizer. Following the text-only RAG setup, we train the retriever using a SupCon loss and the generator with a Cross-Entropy loss.

**Factual Consistency Evaluation.** To evaluate factual consistency, we utilize datasets from FactCC (Kryściński et al., 2019) and QAGS (Wang et al., 2020). For the FactCC benchmark, we combine the publicly available validation (931 samples) and test (503 samples) splits into a single evaluation set. Similarly, we merge the human-annotated QAGS datasets for the CNN/DM and XSUM (Narayan et al., 2018) test sets to form our second evaluation set. For training our consistency model, we use the document-summary pairs from the CNN/DailyMail (CNN/DM) (Nallapati et al., 2016) training dataset. The model is trained for 100 epochs with a batch size of 64, using an Adam optimizer with a learning rate of $10^{-4}$ and a Binary Cross-Entropy loss function.

# D    EVALUATION METRICS

For text classification and factuality consistency, we report the F1 score to assess model performance as either classification label or label for consistencies are available in the test dataset. For both single-modal and multimodal retrieval-augmented tasks, we adopt a suite of evaluation metrics that separately measure the quality of retrieval and generation, following (Lin & Byrne, 2022).

## D.1    RETRIEVAL METRICS

**Recall@K** measures the fraction of ground-truth relevant documents that appear among the top-$K$ retrieved candidates. Let $P$ be the set of ground-truth passage IDs and $\hat{P}_K$ be the set of top-$K$ retrieved passage IDs, then

$$\text{Recall@}K = \frac{|\hat{P}_K \cap P|}{|P|}. \tag{5}$$

This metric provides a fine-grained view of how completely the model recovers all known relevant documents. It is particularly informative when multiple ground-truth passages are available.

**Pseudo-Relevance Recall (PRRecall@K)** is a binary variant of recall. It assigns a score of 1 if at least one ground-truth passage is retrieved among the top-$K$, and 0 otherwise,

$$\text{PRRecall@}K = \begin{cases} 1 & \text{if } \hat{P}_K \cap P \neq \emptyset \\ 0 & \text{otherwise.} \end{cases} \tag{6}$$

PRRecall@K focuses on whether the system retrieves any useful knowledge, regardless of how many relevant passages exist. This is especially useful in open-domain and knowledge-intensive settings such as VQA, where retrieving a single correct passage may be sufficient to produce the correct answer. We utilize PRRecall exclusively for the OKVQA dataset, as its structure often provides multiple relevant documents for a single query.

## D.2    GENERATION METRICS

We evaluate the quality of generated answers using three standard metrics: F1-score, Exact Match (EM), and VQA Score.
**F1-score** measures the token-level overlap between the predicted and ground-truth answers, calculated as the harmonic mean of precision and recall. This offers a more nuanced evaluation than exact match by giving partial credit for answers that share significant wording with the reference.
**Exact Match (EM)** is a strict, binary metric that scores 1 only if the generated answer perfectly

matches a human-annotated reference, and 0 otherwise. It does not account for paraphrasing or partial correctness.

**VQA Score** accounts for consensus among human annotators by giving partial credit to valid but less common answers. The score is calculated as

$$\text{VQA Score}(y, S) = \min\left(\frac{\#S(y)}{3}, 1\right),\tag{7}$$

where $\#S(y)$ denotes the number of annotators who provided the same answer $y$. Together, these metrics provide a comprehensive assessment of the system's ability to retrieve relevant knowledge and generate accurate, human-aligned answers. We use F1-score and EM for single-modal RAG tasks, which typically have a single, longer ground-truth answer. For multimodal RAG, which often has multiple acceptable short answers, we use the VQA score to better reflect answer variability.

**Answer Recall.** To quantify the generator's ability to incorporate retrieved information, we report Answer Recall (also known as Hit Rate). This metric is calculated using substring inclusion on normalized text. Specifically, we apply standard normalization (lowercasing, removing articles, and removing punctuation) to both the ground truth answer and the generated text. A score of 1 is assigned if the normalized ground truth string appears verbatim within the normalized generated response, and 0 otherwise. This ensures the metric credits valid responses embedded in longer generative outputs (e.g., capturing "Paris" within "The capital is Paris") while maintaining strict exact-match criteria for the key terms.

## E    SATPool as a Long-Context Factuality Consistency Evaluator

In addition to creating representations for retrieval and classification, we repurpose SATPool to address a critical challenge in text generation: **factual-consistency evaluation**. This task involves automatically estimating how faithfully a generated text, such as a summary, reflects the facts in its source document. Standard tools for this task include BERTScore (Zhang et al., 2019), which measures token-level semantic overlap using a pre-trained encoder, and FactCC-Eval (Kryściński et al., 2019), a classification model fine-tuned on BERT to predict whether a summary sentence is entailed or contradicted by the source (more details in BERTScore and FactCC are in Appendix G). However, a significant drawback of these approaches is that they inherit the 512-token input limit from their underlying BERT architecture. This forces them to truncate long source documents, meaning factual errors related to content beyond the initial segment may go undetected.

To overcome this, we introduce the **Dense Factuality Consistency Score (DFCScore)**, a model that leverages SATPool to process the entire document without truncation.

**Method**    Given a document $d$ and a candidate summary $s$, DFCScore computes a probability that the pair is factually consistent. Both inputs are processed by a frozen BERT encoder $\mathcal{E}$ to obtain token embeddings. These embeddings are then passed through separate processing heads, $f_d$ and $f_s$, which each consist of a pooling layer followed by a three-layer MLP. For the document head $f_d$, we test both SATPool and a mean-pooling baseline as its pooling component. The final score is then calculated as:

$$\text{DFCScore}(d, s) = \sigma\big(w \cdot \cos(f_d(\mathcal{E}(d)), f_s(\mathcal{E}(s))) + b\big),$$

where $\sigma$ is a sigmoid function, and $w$ and $b$ are learnable scalar parameters for calibration. The model is trained using weak supervision with positive $(d, s)$ pairs from the CNN/DM dataset (Nallapati et al., 2016) and tested on the human-annotated FactCC-D (Kryściński et al., 2019) and QAGS (Wang et al., 2020) benchmarks.

**Hypothesis**    If SATPool truly preserves full-document semantics, the resulting DFCScore should (i) outperform a mean-pooled baseline, and (ii) rival or surpass the truncation-based BERTScore and FactCC-Eval benchmarks, all while remaining length-agnostic.

**Result Analysis**    The results presented in Table 6 validate our hypothesis. First, **DFCScore-SATPool** significantly outperforms DFCScore-Mean on both datasets (e.g., 0.9275 vs. 0.8667 on FactCC-D), confirming that SATPool's holistic representation is superior to simple averaging for

this task. Second, DFCScore-SATPool achieves performance that is highly competitive with the established benchmarks, slightly surpassing BERTScore on FactCC-D and performing on par with both BERTScore and FactCC-Eval on QAGS. These findings demonstrate that by processing the full document, SATPool enables a more robust and accurate measure of factual consistency compared to mean pooling.

Table 6: Experimental results for Factuality Consistency Evaluation (F1 Score). DFCScore with SATPool achieves competitive performance against established benchmarks, while significantly outperforming a mean-pooling baseline.

| F1 Score | FactCC-D | QAGS |
|---|---|---|
| BERTScore | 0.9274 | 0.8087 |
| FactCC-Eval | 0.9183 | 0.8075 |
| DFCScore-Mean | 0.8667 | 0.7845 |
| DFCScore-SATPool | 0.9275 | 0.8073 |

# F  ADDITIONAL EXPERIMENTS

## F.1  IMPACT OF THE NUMBER OF SPECTRAL COMPONENT $k$

Table 7 offers a more detailed breakdown of SATPool's performance by including results for $k = 2$ and $k = 4$, supplementing the main results for $k = 8$ presented in Table 2. The key finding is that while SATPool significantly outperforms the baselines across all tested values of $k$, the optimal number of spectral components is dataset-dependent. For example, smaller values ($k = 2$ and $k = 4$) are most effective for the 20News dataset, whereas a larger value ($k = 8$) is superior for the EURLEX dataset. This suggests that the ideal value of $k$ corresponds to the structural complexity of the documents in a given dataset.

Table 7: Experimental results for text classification task for 20News and EURLEX datasets. **Bold** indicates the best result for each baseline, while underline denotes the second best result.

| F1 score | | 20NewsGroups | | | | EURLEX | | |
|---|---|---|---|---|---|---|---|---|
| | BERT | RoBERTa | Longformer | LongT5 | BERT | RoBERTa | Longformer | LongT5 |
| Baseline (CLS) | 0.5916 | 0.5890 | 0.5798 | 0.2606 | 0.5645 | 0.4975 | 0.4933 | 0.4411 |
| Mean | 0.6223 | 0.6315 | 0.6386 | 0.5637 | 0.6856 | 0.6515 | 0.6644 | 0.6921 |
| Late Chunking | 0.6166 | 0.6115 | 0.6219 | 0.4909 | 0.6763 | 0.6708 | 0.6731 | 0.4960 |
| SATPool  $k = 2$ | 0.6563 | **0.6657** | 0.6548 | **0.6361** | 0.6881 | 0.7222 | 0.7028 | 0.7243 |
| $k = 4$ | **0.6621** | 0.6580 | **0.6634** | 0.6347 | 0.7040 | 0.7107 | 0.7140 | 0.7076 |
| $k = 8$ | 0.6371 | 0.6570 | 0.6450 | 0.6195 | **0.7124** | **0.7266** | **0.7232** | **0.7306** |

## F.2  IMPACT OF LOW-RANK DIMENSION $r$

Given the significant impact of linear attention shown in Figure 6, we further analyze the effect of the low-rank dimension, $r$, used in this attention mechanism. As illustrated in Figure 8, increasing the dimension $r$ leads to consistent performance improvements on the 20News dataset. While a small dimension ($r = 32$) is sufficient to boost performance over the baseline, the results confirm that larger values for $r$ continue to yield better performance by more effectively capturing long-range dependencies. Since the number of added parameters grows linearly with $r$ (totaling $(2K + 4)rd$, where $d$ is the baseline model's hidden dimension), we choose $r = 128$ for our main experiments to maintain a balance between performance and computational efficiency.

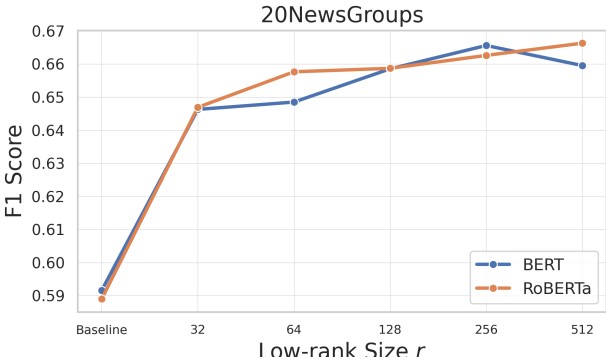

Figure 8: The effect of the low-rank dimension $r$ in the linear attention module on the 20NewsGroups dataset. Performance scales positively with $r$, illustrating the trade-off between model accuracy and the number of parameters.

### F.3 JOINT ANALYSIS OF HYPERPARAMETERS $K$ AND $r$

To address potential concerns regarding interaction effects between hyperparameters and the method's overall sensitivity, we conducted a comprehensive grid search over the number of spectral components ($K$) and the low-rank dimension ($r$). We utilize the RoBERTa backbone on the 20NewsGroups dataset. To ensure the reliability of these findings, we perform three independent runs with different random initializations for each configuration and report the mean F1 Score $\pm$ standard deviation.

The results, presented in Table 8, demonstrate the following characteristics:

**Minimal Interaction Effects.** We observe a broad region of stability rather than a complex dependency where the choice of $K$ strictly dictates the optimal $r$. For instance, setting $K = 2$ yields consistently high performance ($0.6439$–$0.6508$) across the entire range of tested $r$ values.

**Low Sensitivity and Robustness.** The method demonstrates remarkable stability. Across the entire hyperparameter grid (20 configurations), the F1 score varies by a maximum of only $1.6\%$. Furthermore, the standard deviations are consistently low (mostly $< 0.01$), indicating high reproducibility.

**Performance Floor vs. Baseline Ceiling.** Most significantly, the grid search reveals that even the suboptimal configurations of SATPool outperform the best-performing baselines. The lowest score observed in our grid is $0.6378$ (at $K = 4, r = 32$). In contrast, the best baseline performance (RoBERTa with Mean Pooling) reported in Table 2 is $0.6315$. This confirms that users do not need to perform exhaustive tuning to achieve state-of-the-art results; any reasonable hyperparameter choice yields performance superior to standard pooling methods.

Table 8: Joint impact of spectral components ($K$) and low-rank dimension ($r$) on F1 Score (Mean $\pm$ Std) using RoBERTa on 20NewsGroups.

| $K\backslash r$ | 32 | 64 | 128 | 256 | 512 |
|---|---|---|---|---|---|
| 1 | $0.6470 \pm 0.0022$ | $0.6471 \pm 0.0039$ | $0.6465 \pm 0.0055$ | $0.6538 \pm 0.0099$ | $0.6425 \pm 0.0067$ |
| 2 | $0.6482 \pm 0.0085$ | $0.6508 \pm 0.0101$ | $0.6481 \pm 0.0051$ | $0.6439 \pm 0.0042$ | $0.6480 \pm 0.0035$ |
| 4 | $0.6378 \pm 0.0090$ | $0.6502 \pm 0.0088$ | $0.6454 \pm 0.0062$ | $0.6444 \pm 0.0039$ | $0.6381 \pm 0.0023$ |
| 8 | $0.6451 \pm 0.0043$ | $0.6449 \pm 0.0055$ | $0.6479 \pm 0.0056$ | $0.6399 \pm 0.0061$ | $0.6427 \pm 0.0066$ |

### F.4 SENSITIVITY TO CHUNKING STRATEGIES

To evaluate the sensitivity of SATPool to different chunking configurations (chunk size and overlap), we conducted a comprehensive ablation study on the 20NewsGroups dataset. We compare our method against the Mean Pooling baseline using two distinct backbones: BERT (short-context) and Longformer (long-context). The results, summarized in Table 9, reveal the following insights:

**Consistent Superiority.** Across all tested configurations, SATPool consistently outperforms Mean Pooling, regardless of the chunk size or stride strategy employed. For instance, with BERT (Chunk Size 128, 50% Overlap), SATPool achieves an F1 score of 0.6394 compared to Mean Pooling's 0.6020 (+3.7%), confirming that our method's advantage is robust and not an artifact of a specific preprocessing setup.

**Impact of Stride and Overlap.**

- *Short-Context Models (BERT):* A sliding window approach (Stride 0.5) provides a clear benefit. At a chunk size of 256, overlapping chunks improve SATPool's performance from 0.6317 to 0.6450 (+1.3%). This suggests that overlapping helps preserve boundary contexts that SATPool effectively aggregates.
- *Long-Context Models (Longformer):* Conversely, non-overlapping chunks (Stride 1.0) yield superior results. At a chunk size of 1024, non-overlapping performance (0.6618) surpasses overlapping performance (0.6481).

These findings indicate that while sliding-window chunking is beneficial for short-context encoders, it is not strictly necessary for long-context models, where non-overlapping chunks prove to be both highly effective and computationally efficient.

Table 9: Impact of Chunk Size and Stride (Overlap) on F1 Score. SATPool consistently outperforms Mean Pooling across varying chunk sizes and overlap strategies.

| Backbone | Chunk Size | Stride = 0.5 (50% Overlap) | | Stride = 1.0 (No Overlap) | |
|---|---|---|---|---|---|
| | | Mean | SATPool | Mean | SATPool |
| **BERT** | 128 | 0.6020 | **0.6394** | 0.6111 | 0.6224 |
| | 256 | 0.6150 | **0.6450** | 0.6142 | 0.6317 |
| | 512 | 0.6228 | **0.6387** | 0.6229 | 0.6316 |
| **Longformer** | 1024 | 0.6312 | 0.6481 | 0.6306 | **0.6618** |
| | 2048 | 0.6274 | 0.6445 | 0.6301 | **0.6570** |
| | 4096 | 0.6309 | 0.6356 | 0.6284 | **0.6573** |

## F.5 Experimental Results with Large Language Model

In our main experiments, we observed that improvements in retrieval metrics did not always translate linearly to improved generation metrics (e.g., Exact Match) when using standard encoder-decoder models like T5. We hypothesize that this discrepancy might stem from the generator's limited capacity to reason over long contexts or effectively utilize retrieved information, a phenomenon akin to the "lost-in-the-middle" problem.

To test this hypothesis and demonstrate the flexibility of SATPool with modern generative backbones, we replace the T5 generator with a Large Language Model (LLM), specifically Qwen2.5-3B-Instruct. We utilize the same retrieval pipeline but feed the aggregated context to the LLM for the final answer generation. The results (Table 11), as detailed in the following subsection, indicate that a stronger generator significantly benefits from the high-fidelity retrieval provided by SATPool.

## F.6 Comparison with Stronger Hierarchical Baselines

To rigorously evaluate SATPool's efficacy, we compare it against established hierarchical transformer baselines: TrLDC (Transformer-based Long Document Classification) Dai et al. (2022) and PARADE Li et al. (2023). These methods represent a significant class of document encoding strategies that aggregate representations from local chunks.

**Text Classification Results.** Table 10 presents the performance on the 20NewsGroups dataset. SATPool consistently outperforms both PARADE and TrLDC across BERT and RoBERTa backbones. Notably, the hierarchical baselines often underperform simple Mean Pooling in this setting.

**RAG Performance.** We further evaluate these methods on the Natural Questions (NQ) dataset. To ensure robust evaluation, we employed bootstrapping on the full test set, reporting the mean and

Table 10: Comparison with hierarchical baselines on 20NewsGroups (F1 Score).

| Method | F1 (BERT) | F1 (RoBERTa) |
|--------|-----------|--------------|
| CLS (Truncation) | 0.5916 | 0.5890 |
| Mean Pooling | 0.6223 | 0.6315 |
| Late Chunking | 0.6166 | 0.6115 |
| PARADE | 0.5819 | 0.6070 |
| TrLDC | 0.5754 | 0.5705 |
| **SATPool (Ours)** | **0.6371** | **0.6570** |

standard deviation. As shown in Table 11, SATPool achieves the highest performance across all metrics, including Answer Recall generated by `Qwen2.5-3B-Instruct`.

Table 11: RAG Performance on Natural Questions (Full Set). Results are reported as Mean $\pm$ Standard Deviation derived from 1,000 bootstrap resamples, ensuring statistical robustness. SATPool consistently outperforms hierarchical baselines.

| Method | Recall@1 | Recall@5 | Answer Recall |
|--------|----------|----------|---------------|
| *Baselines* | | | |
| No RAG (Generator Only) | - | - | $0.1861 \pm 0.0072$ |
| CLS (Truncation) | $0.0545 \pm 0.0043$ | $0.1755 \pm 0.0073$ | $0.4195 \pm 0.0090$ |
| MaxP | $0.0619 \pm 0.0048$ | $0.2657 \pm 0.0086$ | $0.4984 \pm 0.0093$ |
| Mean Pooling | $0.1969 \pm 0.0095$ | $0.4149 \pm 0.0125$ | $0.5795 \pm 0.0237$ |
| Late Chunking | $0.1894 \pm 0.0124$ | $0.4175 \pm 0.0138$ | $0.5799 \pm 0.0137$ |
| *Hierarchical Aggregators* | | | |
| PARADE | $0.1314 \pm 0.0068$ | $0.2972 \pm 0.0088$ | $0.5019 \pm 0.0093$ |
| TrLDC | $0.1275 \pm 0.0065$ | $0.2697 \pm 0.0089$ | $0.5033 \pm 0.0093$ |
| **SATPool (Ours)** | $\mathbf{0.2371 \pm 0.0083}$ | $\mathbf{0.5037 \pm 0.0096}$ | $\mathbf{0.6302 \pm 0.0093}$ |

**Analysis: Pool-then-Attend vs. Attend-then-Pool.** The underperformance of hierarchical methods like TrLDC and PARADE in our frozen-encoder setup highlights a critical architectural distinction. These methods follow a *"Pool-then-Attend"* paradigm: they first compress each chunk into a single `[CLS]` vector before aggregation. When using frozen encoders not fine-tuned for the specific downstream task, the generic `[CLS]` token often captures suboptimal representations of local chunks, creating an early information bottleneck.

In contrast, SATPool follows an *"Attend-then-Pool"* paradigm. Our Stage 1 (Linear Attention) operates on the *full sequence of tokens* across all chunks. This allows the model to discover and weigh global semantic relationships at the high-fidelity token level *before* any compression occurs in Stage 2 (STC), thereby preserving significantly more task-relevant information.

### F.7 HYBRID FINE-GRAINED MATCHING: SPECTRAL COMPONENT INTERACTION

We further investigate the potential of extending STC to support hybrid fine-grained matching, inspired by late-interaction paradigms (e.g., ColBERT). While standard token-level interaction is prohibitively expensive for long documents, our STC framework allows for a mathematically elegant **Spectral Component Interaction (SCI)** that operates on the $2K + 1$ coefficient vectors.

**Methodology.** We adapt our pre-trained SATPool model by fine-tuning a lightweight scoring head for this interaction task. Instead of compressing the document to a single vector, we represent it as a set of $2K + 1$ spectral components (the Mean vector plus Sine/Cosine vectors). We treat these components as an ensemble of structural views. The relevance score is computed by averaging the fine-grained similarities between the query vector $q$ and each spectral component $\mathbf{v}_k$:

$$S_{SCI}(q, D) = \frac{1}{2K+1} \sum_{k=0}^{2K} (q^\top \mathbf{v}_k) \tag{8}$$

This formulation allows the model to evaluate relevance against specific structural frequencies (e.g., global mean vs. high-frequency transitions) before aggregating the signals.

**Results.** We evaluate this hybrid extension on the NQ dataset using Qwen2.5-3B-Instruct for generation. Table 12 compares the performance of standard Dense SATPool against SCI configurations.

Table 12: Comparison of standard Dense SATPool vs. Spectral Component Interaction (SCI). Increasing structural resolution ($K$) improves precision at the cost of retrieval speed.

| Method | Components $(2K+1)$ | Rel. Time | Recall@5 | Answer Recall |
|---|---|---|---|---|
| **SATPool (Dense)** | **1** | **1x** | $0.4842 \pm 0.0190$ | $0.6288 \pm 0.0284$ |
| SATPool (SCI, $K=4$) | 9 | $\sim 9$x | $0.4701 \pm 0.0035$ | $0.6312 \pm 0.0132$ |
| SATPool (SCI, $K=8$) | 17 | $\sim 17$x | $\mathbf{0.5266 \pm 0.0060}$ | $\mathbf{0.6604 \pm 0.0073}$ |

The results demonstrate that increasing structural resolution ($K=8$) significantly improves both retrieval quality (**+4.2%** Recall@5) and downstream generation (**+3.1%** Answer Recall). This confirms that the spectral components extracted by SATPool contain rich, fine-grained signals. However, for large-scale applications requiring maximum efficiency, the standard Dense SATPool remains the optimal choice, offering competitive performance with significantly faster retrieval.

### F.8 JUSTIFICATION OF FIXED SPECTRAL BASES

To justify our design choice of fixed sinusoidal bases over learnable bases or Principal Component Analysis (PCA), we compare SATPool against two rigorous baselines:

1. **Learned Bases (Linear Interpolation):** To handle variable lengths robustly, we implement learned positional weights $W_{pos} \in \mathbb{R}^{N_{basis} \times 2048}$ and apply linear interpolation during the forward pass to match the input document length $L$. This isolates structural overfitting from simple length mismatch issues.

2. **PCA Bases (Dynamic SVD):** We compute Principal Components on-the-fly for each document instance, projecting tokens onto the top-$N_{basis}$ eigenvectors of the covariance matrix. This serves as an ideal "Bag-of-Words" baseline that captures dominant semantic topics but discards sequential structure.

We evaluated these methods on the Natural Questions (NQ) dataset under two settings: **Standard** and **Permuted** (shuffling document chunks to destroy narrative structure).

Table 13: RAG Performance on NQ (Standard vs. Permuted Test Set). Fixed spectral bases demonstrate superior robustness compared to Learned or PCA bases.

| Method | Standard Test Set | | | Permuted Test Set | | |
|---|---|---|---|---|---|---|
| | **R@1** | **R@5** | **Ans. Recall** | **R@1** | **R@5** | **Ans. Recall** |
| SATPool w/ Learned | 0.2356 | 0.4629 | 0.6133 | 0.1776 | 0.3970 | 0.5689 |
| SATPool w/ PCA | **0.2386** | 0.5009 | 0.6274 | 0.2089 | 0.4561 | 0.6176 |
| **SATPool (Spectral)** | 0.2371 | **0.5037** | **0.6302** | **0.2203** | **0.4821** | **0.6274** |

**Analysis.** As shown in Table 13, **Learned Bases** suffer a significant performance drop on permuted documents (Recall@5 drops from 0.4629 to 0.3970), confirming that learned weights tend to overfit to rigid absolute positions rather than relative structure. **PCA Bases** perform well on the standard set due to strong keyword matching but fail to match SATPool's robustness in the permuted setting. **SATPool (Spectral)** achieves the highest performance and stability, confirming that fixed spectral bases provide a generalized, length-agnostic structural prior that captures global context without overfitting to specific positional patterns.

# G DETAILS OF COMPARISON METHODS

## G.1 LATE CHUNKING (GÜNTHER ET AL., 2024)

The Late Chunking (Günther et al., 2024) method critiques standard fixed-length chunking, arguing that its arbitrary start and end points can disrupt the document's semantic context. To address this, Late Chunking proposes segmenting the document along natural semantic boundaries, specifically sentences, which are identified by punctuation.

The process begins by generating an embedding for the entire document while simultaneously recording the token indices corresponding to sentence boundaries. This full sequence of token embeddings is then divided into context-aware chunks using the recorded indices. Finally, each chunk's embedding is calculated via mean pooling, and these are averaged together to produce the single, final representation for the document.

## G.2 MAXP (DAI & CALLAN, 2019)

The MaxP (Maximum Passage Similarity) method, used in LongRAG (Jiang et al., 2024), is a pooling strategy for determining a document's relevance to a query. Unlike methods that aggregate information from all chunks, MaxP's approach is selective, basing the document's entire relevance on its single most similar passage to the query. The process involves computing a similarity score between the query and every chunk of the document. The final relevance score is then defined as the single maximum score from this set of comparisons. In essence, the document's value is judged by its most relevant part, assuming that a single strong match is sufficient to make the entire document useful.

## G.3 BERTSCORE (ZHANG ET AL., 2019)

BERTScore is an evaluation metric designed to judge the quality of a machine-generated sentence by comparing its semantic meaning to a correct, reference sentence (x), rather than just matching keywords. The process begins by using a language model, BERT, to understand the contextual meaning of each word in both the candidate ($\hat{x}$) and reference sentences. Following this, the metric computes a similarity score for every possible pairing of words and employs a simple greedy matching strategy. This strategy works by taking each word from one sentence and pairing it with the single, most similar-meaning word from the other sentence, based on their cosine similarity. These best-match scores are then used to calculate two key metrics. Recall measures if the generated text captured all the ideas from the reference; it does this by looking at each word in the reference sentence and finding its best match in the candidate. Precision, conversely, measures if all the words in the generated text were relevant; it looks at each word in the candidate sentence and finds its best match in the reference. These concepts are formally defined as:

$$R_{BERT} = \frac{1}{|x|} \sum_{x_i \in x} \max_{\hat{x}_j \in \hat{x}} x_i^\top \hat{x}_j \quad P_{BERT} = \frac{1}{|\hat{x}|} \sum_{\hat{x}_j \in \hat{x}} \max_{x_i \in x} x_i^\top \hat{x}_j$$

A final F1-score ($F_{BERT}$) is then calculated as the harmonic mean of Precision and Recall to provide a single, balanced measure.

## G.4 FACT-CC (KRYŚCIŃSKI ET AL., 2019)

FactCC is a weakly-supervised, model-based approach for verifying the factual consistency of an abstractive summary against a source document. The core of the FactCC system is a BERT-based classifier fine-tuned on a large, artificially generated dataset because no large-scale, human-annotated corpus for this specific task exists. The training data is created by first extracting sentences from source documents and then applying a series of rule-based textual transformations to generate both positive (factually consistent) and negative (inconsistent) examples. Semantically invariant transformations, such as paraphrasing via back-translation, are used to create consistent claims. Semantically variant transformations are used to create inconsistent claims and include negation, pronoun swapping, and randomly swapping entities and numbers with others found in the source document. The model architecture takes the full source document and a single summary sentence as input and performs a binary classification (CONSISTENT/INCONSISTENT) using the `[CLS]` token representation.

### G.5   PARADE LI ET AL. (2023)

PARADE (Passage Representation Aggregation for Document Reranking) is a hierarchical method originally designed to aggregate passage representations for document ranking tasks. The method operates by first encoding individual document chunks independently using a standard BERT encoder to obtain the `[CLS]` token embedding for each chunk. These chunk-level representations are then fed into a Transformer-based aggregator layer to model the interactions between passages and produce a final document relevance score. In our experiments, we adapt this aggregation strategy to pool chunk representations into a unified document embedding on top of the frozen backbone.

### G.6   TRLDC DAI ET AL. (2022)

TrLDC (Transformer-based Long Document Classification) proposes a hierarchical architecture specifically optimized for long document classification. Similar to PARADE, it begins by segmenting the document into chunks and encoding each to obtain a `[CLS]` representation. To better capture the document's structure, TrLDC adds learnable segment position embeddings to these chunk vectors. The resulting sequence of chunk embeddings is processed by a Transformer encoder to capture inter-chunk dependencies, followed by a Max Pooling operation to derive the final global document representation.

## H   COMPUTATIONAL EFFICIENCY ANALYSIS

### H.1   DETAILED RUNTIME AND FLOPS ANALYSIS

We report the full computational benchmarks for SATPool against baselines in Table 14. The experiment involved processing a document of length 19,502 tokens (approx. 91.5k characters) using a BERT backbone.

Table 14: Wall-Clock Speed and Computational Cost (BERT Backbone). SATPool achieves near-baseline speed while significantly outperforming Long-Context models.

| Method | Time (ms) | FLOPs | Relative Speed |
|---|---|---|---|
| *Baselines* | | | |
| Truncation (CLS) | 447.37 | 6.05e+08 | 3.0x |
| MaxP | 989.76 | 4.66e+10 | 1.3x |
| Mean Pooling | 1333.84 | 4.65e+10 | 1.0x (Baseline) |
| *Advanced Aggregators* | | | |
| PARADE Li et al. (2023) | 1024.73 | 4.67e+10 | 1.3x |
| TrLDC Dai et al. (2022) | 1020.18 | 4.67e+10 | 1.3x |
| Late Chunking | 1657.78 | 4.65e+10 | 0.8x |
| **Longformer (CLS)** | 2356.94 | 4.59e+10 | 0.57x |
| **SATPool (Ours)** | **1347.72** | **5.26e+10** | **∼1.0x** |

**Hyperparameter Impact.** To assess the cost of our hyperparameters, we measured performance across a grid of $K$ (spectral components) and $r$ (low-rank dimension).

- $K$ **is Free:** Increasing $K$ from 2 to 8 results in almost identical FLOPs (e.g., 4.95e10 constant for $r = 64$). This confirms that increasing structural resolution incurs zero penalty during the heavy attention computation.
- $r$ **Controls Trade-off:** The cost scales linearly with $r$. However, even tripling the dimension from $r = 64$ to $r = 256$ results in only a modest $\sim 20\%$ increase in FLOPs, offering a flexible dial for efficiency vs. performance.

### H.2   MEMORY BENCHMARK

We measured the **Peak CUDA Memory** allocated during the forward pass of a single long document (19,502 tokens) on an NVIDIA RTX A5000 (24GB). We report the "Total System Memory," which

includes both the Retriever (Encoder + Pooling) and the Generator (Qwen2.5-3B), to demonstrate the real-world impact.

Table 15: Peak Memory Usage (Single Document Inference). SATPool introduces negligible overhead relative to the full system memory.

| Method | Total Sys Mem (MB) | Overhead vs. CLS (MB) | % of CLS Baseline |
|---|---|---|---|
| Truncation (CLS) | 11,775.56 | - | 100.00% |
| MaxP | 11,800.31 | +24.75 | 100.21% |
| PARADE | 11,805.28 | +29.72 | 100.25% |
| TrLDC | 11,805.43 | +29.87 | 100.25% |
| Mean Pooling | 11,856.97 | +81.41 | 100.69% |
| Late Chunking | 11,858.99 | +83.43 | 100.71% |
| **SATPool (Ours)** | **11,925.24** | **+149.68** | **101.27%** |

The results show that the heaviest method (SATPool) consumes only $\sim 150$MB more than the lightest baseline. Given standard GPU VRAM capacities (24GB+), this overhead is trivial, confirming that the generator is the primary memory constraint in long-context RAG systems.

## H.3 THEORETICAL ANALYSIS

### H.3.1 DENSE LOCAL VS. SPARSE GLOBAL ATTENTION

A fundamental question arises regarding the theoretical upper bound of pooling methods compared to dedicated long-context models: Can a model that encodes chunks in isolation (like SATPool) ever rival a model that accesses global context during encoding (like Longformer)?

Our results suggest that the answer lies in the trade-off between attention density and context length.

- **Sparse Global Attention:** Architectures like Longformer (Beltagy et al., 2020) rely on sparse attention patterns (e.g., sliding windows) to manage quadratic complexity. While they access global context, they sacrifice dense token-to-token connectivity, potentially missing fine-grained dependencies.
- **Dense Local + Global Pooling:** SATPool utilizes standard encoders (e.g., BERT) that preserve full, dense attention within each chunk. Stage 1 then recovers global connectivity via Linear Attention.

Empirically, our results on EURLEX (Table 2) demonstrate that preserving high-fidelity local representations combined with global relational aggregation (SATPool) often outperforms the sparse-attention approximations used by native long-context architectures. This indicates that the "information loss" from chunking can be less detrimental than the "fidelity loss" inherent in sparse attention mechanisms.

### H.3.2 INDUCTIVE BIAS AND SEMANTIC TRAJECTORIES

**The Necessity of Inductive Bias.** Standard mean pooling is a permutation-invariant operation; it creates a many-to-one mapping where structurally distinct sequences (e.g., swapping the introduction and conclusion) produce identical vectors. Consequently, structural signals are lost before reaching the classifier, regardless of downstream training. SATPool addresses this by introducing a necessary inductive bias: it modulates embeddings with spectral bases *before* aggregation. This renders the operation permutation-sensitive, explicitly encoding global positions into the feature magnitude and preserving the document's narrative flow.

**Modeling Semantic Trajectories.** We view a sequence of token embeddings not as a static bag of meanings, but as a *semantic trajectory* $H \in \mathbb{R}^{L \times d}$ evolving over time. Spectral decomposition analyzes the rate of change of this trajectory. Low-frequency components capture the global narrative arc (e.g., the shift from problem to solution), while high-frequency components often correspond to local noise. By learning coefficient vectors for these bases, SATPool acts as a trainable filter, identifying the specific structural frequencies that contain the most discriminative information for the downstream task.

## I  THE USE OF LARGE LANGUAGE MODELS (LLMs)

We employed an LLM as a writing assistant during the preparation of this manuscript. The LLM's role was strictly limited to improving the grammar, clarity, and phrasing of existing text. Each modification suggested by the model was carefully reviewed by the authors to ensure the original scientific meaning and technical details remained accurate and unchanged. The LLM did not contribute to the core research ideation, methodology, experimental results, or conclusions presented in this paper. The authors take full responsibility for all content in this manuscript.

