# OpenReview forum: "Beyond Chunking: Efficient Global Pooling for Holistic Long-Document Representation"
_ICLR.cc/2026/Conference — Submitted to ICLR 2026_

### Official Review · Reviewer_75yK · 2025-10-30

**Soundness:** 3
**Presentation:** 3
**Contribution:** 2
**Rating:** 2
**Confidence:** 4

**Summary:**

Encoding and representing long documents with Transformer encoder-based models, such as BERT, remains a significant challenge. A common solution is to split long contexts into smaller chunks, encode each chunk into an embedding, and then pool these embeddings to obtain a final document representation. This paper introduces a novel and efficient pooling strategy within this framework, demonstrating substantial improvements over existing pooling methods in both performance and parameter efficiency.

Given a set of chunk embeddings for a long document, the proposed method first applies linear attention to all embeddings, leveraging techniques from the Performer architecture. It then introduces a discrete-Fourier-transform-like approach, incorporating sine and cosine coefficients into the embedding representations. This projects the original token embeddings onto a set of K bases with varying frequencies. The final document representation is formed by concatenating these frequency-based bases with the result of standard mean pooling.

The authors conduct extensive experiments on four long-context datasets, comparing their method against three to four traditional pooling strategies. Results indicate that the proposed approach consistently outperforms others on metrics such as recall@K, F1, and exact match. Additionally, ablation studies reveal that the method achieves better performance with fewer parameters compared to mean pooling.

**Strengths:**

1. The introduction of sine and cosine coefficients into the representation is a novel and interesting idea that helps preserve structural information within the document. The experiments convincingly demonstrate the feasibility of this approach in both retrieval and certain question answering tasks.
2. The proposed method is model-agnostic and parameter efficient. While it is particularly effective for long-context encoding, it is also broadly applicable to a wider range of use cases beyond just long document processing.

**Weaknesses:**

1. The overall contribution of the paper is limited. There is no evidence that the proposed pooling strategy improves end-to-end question answering performance or achieves state-of-the-art (SOTA) or near-SOTA results on the selected datasets. It would be informative to see how the proposed approach performs when integrated with existing SOTA methods.
2. The evaluation does not include comparisons with strong baselines. The use of mean pooling and MaxP as baselines is rather simplistic, and the Late Chunking approach does not appear to be a robust alternative, either. The results would be more convincing if the proposed method were compared against established SOTA approaches.

**Questions:**

n/a

---

> ### Author Response · Authors · 2025-11-21
>
> ## Weakness 1: On End-to-End Performance (Scope & Validation)
>
> We thank the reviewer for this comment. We would like to clarify the scope and provide additional validation.
>
> 1.  **Scope of Contribution:** Our work focuses on "better dense representation and retrieval," not on optimizing generation architectures. We respectfully posit that optimizing the generator itself or analyzing generator-side failures in depth is outside the scope of this paper, which aims to provide a robust plug-and-play pooling module for long-document encoding.
> 2.  **Validation with SOTA LLM:** However, we accept the reviewer's point that demonstrating downstream utility with a capable model is informative. We replaced the T5 generator with **Qwen2.5-3B-Instruct**. To ensure robustness, we report the **Mean and Standard Deviation across 4 random data subsets**.
>
>
> We are confident in our method's universality and welcome any suggestions for specific generator models the reviewer would like to see tested.
>
>
> - **RAG Performance (Natural Questions)**
>
> *Note: We report Answer Recall using Qwen2.5-3B-Instruct as the generator.*
>
> | Method | Recall@1 | Recall@5 | Answer Recall |
> | :--- | :--- | :--- | :--- |
> | No RAG | - | - | 0.1915 ± 0.0251 |
> | CLS (Truncation) | 0.0553 ± 0.0112 | 0.1766 ± 0.0147 | 0.4147 ± 0.0216 |
> | MaxP | 0.0587 ± 0.0028 | 0.2596 ± 0.0117 | 0.5052 ± 0.0323 |
> | Mean Pooling | 0.1969 ± 0.0095 | 0.4149 ± 0.0125 | 0.5795 ± 0.0237 |
> | PARADE | 0.1348 ± 0.0098 | 0.3022 ± 0.0204 | 0.5073 ± 0.0248 |
> | TrLDC | 0.1312 ± 0.0092 | 0.2709 ± 0.0169 | 0.4958 ± 0.0058 |
> | Late Chunking | 0.1894 ± 0.0124 | 0.4175 ± 0.0138 | 0.5799 ± 0.0137 |
> | **SATPool (Ours)** | **0.2449 ± 0.0048** | **0.4842 ± 0.0190** | **0.6288 ± 0.0284** |
>
>
> ## Weakness 2:  Comparison with Stronger Baselines
>
> We agree that Mean/MaxP are simplistic. To provide a rigorous comparison against established SOTA approaches, we implemented **PARADE** and **TrLDC** as pooling modules on top of the same frozen encoders.
>
> * **RAG Results:** Please refer to the table in **Response to W1** above. SATPool significantly outperforms both PARADE and TrLDC in retrieval and downstream generation.
> * **Text Classification Results:** As shown below, SATPool consistently outperforms these baselines on the 20NewsGroups dataset.
>
> - **Text Classification Performance (20NewsGroups)**
>
> | Method | F1 score (BERT) | F1 score (RoBERTa) |
> | :--- | :--- | :--- |
> | CLS (Truncation) | 0.5916 | 0.5890 |
> | Mean Pooling | 0.6223 | 0.6315 |
> | PARADE | 0.5819 | 0.6070 |
> | TrLDC | 0.5754 | 0.5705 |
> | Late Chunking | 0.6166 | 0.6115 |
> | **SATPool (Ours)** | **0.6371** | **0.6570** |
>
>
> **Analysis:**
> We attribute the lower performance of PARADE and TrLDC to the "Frozen Bottleneck". These architectures typically rely on end-to-end fine-tuning to compress information into the `[CLS]` token. When used with a frozen encoder (plug-and-play setting), the `[CLS]` token fails to capture sufficient local context. SATPool avoids this by attending to the full token sequence (Attend-then-Pool), enabling robust performance even with frozen backbones.
>
>
> We are fully committed to rigorous evaluation. We welcome any additional suggestions for comparison methods that the reviewer believes would further strengthen the validation of our plug-and-play approach. If there are other specific aggregation modules the reviewer would like to see tested, we are more than willing to conduct those experiments.
>
>
> [1] Li, et. al. PARADE: Passage Representation Aggregation for Document Reranking (2021)
> [2] Dai, et. al. Revisiting Transformer-based Models for Long Document Classification (2022)

---

> > ### Comment · Reviewer_75yK · 2025-11-27
> >
> > Thank you for submitting the supplemental experiments and analysis. After carefully reviewing the updated results and engaging in productive discussions with fellow reviewers, I am now more optimistic about the contribution of this work.
> > **Key Points:**
> >
> > 1. The observed improvement in retrieval, resulting in higher answer recall with Qwen-3B-Instruct, is noteworthy. However, the F1/EM results in Tables 3 and 4 suggest that SATPool provides limited benefit for most QA tasks, with its primary advantage confined to retrieval. This limitation reduces the overall impact of SATPool. If the "needle-in-a-haystack" issue is due to the generator's quality, I recommend applying Qwen to the other three QA tasks as well. A holistic long-document representation should, in principle, extend beyond retrieval. Additionally, I did not find a clear statement in the manuscript indicating that the work is solely focused on the retrieval problem.
> >
> > 2. The new ablation studies on parameters K and r are valuable for future adoption, offering further insight into the computational efficiency. As reviewer A1Ka suggested, an analysis from the memory perspective would further enhance the work.
> >
> > 3. The addition of two further baselines (PARADE, TrLDC) strengthens the credibility of the results.
> >
> >
> > Based on the supplemental evidence, I am raising my final score from 2 to 4, primarily due to the first point. I appreciate the core idea and the innovation and would like to see it applied to broader use cases. However, the current script requires improvement, which prevents me from increasing the score further.

---

> > > ### Author Response · Authors · 2025-12-02
> > >
> > > We sincerely thank the reviewer for raising the score and for the constructive dialogue. We are encouraged that the additional baselines (PARADE, TrLDC) and the Qwen experiments have strengthened your view of our work.
> > >
> > > We appreciate your candid feedback regarding the manuscript's clarity and scope. We are fully committed to revising the final script to address your remaining concerns.
> > >
> > > **1. On Generator Quality and Scope Definition (Point 1)**
> > >
> > > We agree with the reviewer’s assessment that the T5 results in Tables 3 and 4 obscured the full value of SATPool. However, we would like to clarify our position regarding the scope of the work:
> > >
> > > * **Representation-Centric, Not Just RAG-Centric**: We respectfully clarify that our work is not solely focused on RAG tasks, but is rather **"representation-centric."** Our core contribution is a pooling mechanism that creates a dense, holistic embedding for long documents. This is why our evaluation spans four distinct tasks: Classification, RAG, Multimodal RAG, and Factuality Consistency Evaluation. Limiting the definition to RAG would undercut our contributions in classification (Table 2) and factuality (Table 6), where generation is not the primary bottleneck.
> > > * **Isolating Representation Quality:** We view the high **Answer Recall** with Qwen (0.6288 vs 0.4147 baseline) as definitive proof that SATPool successfully encodes the necessary information into the dense representation. The failure of weaker generators (like T5) to utilize this representation is a "needle-in-a-haystack" generator limitation, which is a separate issue from the quality of the pooling layer. Focusing on optimizing the generator would obscure our specific contribution to vector representation.
> > > * **Commitment to LLM Evaluation:** We fully accept the recommendation to demonstrate this "representation quality" across all QA tasks to prove the gains are not accidental. In the final version, we commit to extending the LLM evaluation to the remaining datasets:
> > >     * **HotpotQA:** Using Qwen2.5-3B-Instruct.
> > >     * **OKVQA & EVQA (Multimodal):** Using Qwen2.5-VL-3B-Instruct to handle visual inputs.
> > > * **Manuscript Revision:** We will revise the Introduction and Problem Definition to be more explicit that our goal is **"dense, holistic representation learning"** which serves as a robust upstream foundation for downstream tasks (whether classification or generation), ensuring the distinction between "pooling quality" and "generator capability" is clear.
> > >
> > > **2. On Memory Analysis and Baselines (Points 2 & 3)**
> > >
> > > We are glad the $K/r$ ablation studies and the new hierarchical baselines (PARADE, TrLDC) enhanced the credibility of our results. As suggested, we have integrated the detailed memory analysis (discussed in the rebuttal to Reviewer A1Ka) into **Appendix H** of the revised paper to provide a complete picture of computational efficiency.
> > >
> > > **Conclusion**
> > >
> > > We are grateful for your recognition of SATPool's innovation. We believe that by clarifying our scope as "representation-centric" and verifying the representation quality with capable LLMs (Qwen/Qwen-VL) across all QA tasks, the final manuscript will clearly demonstrate the broader impact of this work.

---

### Official Review · Reviewer_A1Ka · 2025-10-30

**Soundness:** 2
**Presentation:** 3
**Contribution:** 2
**Rating:** 4
**Confidence:** 4

**Summary:**

The paper presents Spectral Attention Token Pooling (SATPool), an encoder-agnostic module to produce a single fixed-size, holistic vector for arbitrarily long documents. SATPool first applies an efficient Performer-style linear attention over the concatenated chunk embeddings to produce attention-reweighted token representations, then applies Spectral Token Compression (STC): it concatenates the mean embedding and the projections of the sequence onto K sine/cosine bases (i.e., spectral coefficients) to form a compact vector which is linearly projected to the final document embedding. Experiments on long-document classification (20News, EURLEX), RAG (NQ, HotPotQA), multimodal RAG (OKVQA, EVQA) and factuality-consistency tasks show consistent gains over standard pooling, MaxP, and a Late-Chunking baseline.

**Strengths:**

1. **Clear, intuitive two-stage idea.** Combining a linear global attention to collect cross-chunk interactions with a spectral pooling that explicitly encodes coarse positional structure is conceptually neat and interpretable (reconstructed signals / heatmaps). The STC visualization helps build intuition.
2. **Empirical gains across multiple tasks.** The paper reports consistent improvements in retrieval and classification metrics across several datasets and encoders (BERT, RoBERTa, GTR-T5, CLIP-DPR), supporting generality.

**Weaknesses:**

1. **Limited theoretical justification for STC.** STC uses fixed sinusoidal bases and concatenated coefficient vectors; while empirically effective, the paper lacks a deeper theoretical analysis on what spectral components capture in semantic terms and when such bases are sufficient (vs. learned bases or other positional encodings). The robustness test of paragraph reordering is helpful but not fully convincing as a general explanation.
2. **Efficiency / runtime / memory evaluation is shallow.** The paper motivates linear attention for scalability, but lacks a thorough wall-clock / memory benchmarking comparing SATPool (Performer + STC) to alternatives (late-chunking, Longformer, LongT5) under the same hardware and realistic document lengths. Practical adoption will depend heavily on throughput/latency tradeoffs for indexing and retrieval.
3. **Reproducibility choices and dataset sampling.** For some RAG experiments authors train on subsets (e.g., 20k training, 1k test) "for efficiency" — this raises the question whether gains hold at full scale and whether hyper-parameters are overfit to small subsets. More details on hyper-parameter search ranges and robustness across seeds would strengthen claims.
4. **Generation-side impact remains unclear.** While retrieval metrics improve, generation EM/F1 does not always follow; the paper notes the “needle-in-a-haystack” issue but stops short of analyzing why improved retrieval representation fails to translate to generation improvements (e.g., quality of retrieved passages, summary compression step, generator capacity). Additional analysis (cases where retrieval improved but generator still failed) is needed.
5. **Comparisons to latest baselines missing or limited.** The paper compares to mean, MaxP, Late Chunking and some long-context models, but does not include recent SOTA long-document dense retrievers (e.g., MemLong, Longtriever variants with similar experimental settings) beyond a subset of references; stronger, consistent baselines would contextualize gains.

**Questions:**

1. How sensitive is SATPool to the chunking strategy (chunk size, overlap / stride)? Is sliding-window chunking necessary, or do non-overlapping chunks produce similar H? Please report an ablation.
2. Please provide wall-clock throughput / memory comparison for SATPool vs. Late Chunking and Longformer-based encoders on a realistic document corpus. What are r and K choices in practice for long web-scale corpora?
3. Why use fixed sinusoidal bases rather than direct learnable data-driven basis? Or have you tried learned frequency bases or PCA/low-rank bases over token embeddings? Do you have some empirical comparison?
4. For the RAG experiments, when retrieval improves but EM doesn't, can you provide qualitative examples and an analysis that separates (a) retrieval error, (b) summarizer failure (if any in RAG system), and (c) generator misuse of retrieved context? This would illuminate the "needle-in-a-haystack" discussion more clearly.
5. Could STC be combined with late-interaction (dot-product over chunk-level tokens) approaches (i.e., hybrid fine-grained matching)? Any experiments or discussion is welcomed.

---

> ### Author Response · Authors · 2025-11-21
>
> ## Weakness 1 and Question 3: Theoretical Justification and Choice of Spectral Bases
>
>
> We thank the reviewer for pointing out the need for a deeper theoretical analysis of what spectral components capture. We address the theoretical roles of these components and the justification for fixed bases below.
>
> **1. Theoretical Interpretation: Semantics in the Frequency Domain**.
> The reviewer notes a lack of analysis on what spectral components capture in semantic terms. As briefly discussed in Section 4.3, our framework treats the sequence of token embeddings as a multi-dimensional discrete signal.
>
> * **Global Semantics (Mean Vector):** The Mean vector in our architecture captures the static, global semantic content of the document. It represents the "center" of the document's meaning, averaging out all positional variations.
> * **Global Structure (Spectral Components):** The spectral components (K) capture the global structure or topic progression. In a document signal, low-frequency variations correspond to slow, global shifts in topics, such as the transition from "Introduction" to "Methodology" to "Conclusion."
>
> **2. Why Fixed Bases? (Inductive Bias and Extrapolation)**.
> The reviewer asks why we use fixed sinusoidal bases rather than learned ones. We chose fixed bases to impose a strong inductive bias for sequentiality, a principle grounded in the foundational Transformer architecture (Vaswani et al., "Attention Is All You Need", 2017).
>
> * **Performance Parity:** As noted in the original Transformer paper (Vaswani et al., 2017), experiments comparing learned positional embeddings versus fixed sinusoidal ones showed that "the two versions produced nearly identical results."
> * **Extrapolation to Unseen Lengths:** Given that performance is identical, we selected the fixed sinusoidal version because, as Vaswani et al. note, it allows the model to "extrapolate to sequence lengths longer than the ones encountered during training." This is critical for our goal of a robust module that handles various or unseen document lengths effectively without the risk of overfitting to specific positions.
>
> **3. The Limitation of PCA (Permutation Invariance)**.
> The reviewer suggests "PCA/low-rank bases" as an alternative. While data-driven, PCA is theoretically ill-suited for this task due to Permutation Invariance.
>
> * **Bag-of-Vectors vs. Sequence:** PCA calculates basis vectors based on the covariance of the token set. Because covariance depends only on the values of the tokens and not their order, shuffling the document's chunks would yield the same PCA basis. It essentially provides a "better Mean" but fails to capture sequential structure.
> * **Structure-Aware:** STC is explicitly designed to capture positional dynamics. It projects embeddings onto a basis inherently tied to sequence position.
> * **Empirical Evidence:** Our Mean pooling baseline serves as the fundamental static, data-driven approximation. As shown in Figure 6, the structure-aware SATPool (Mean + STC) significantly outperforms this data-driven baseline. This confirms that the structural signal captured by our fixed basis provides critical information that data-driven approaches miss.
>
> **4. Trainable Coefficients**.
> Finally, we emphasize that while the basis functions are fixed, the module is fully trainable. The model learns the projection coefficients (importance weights) via the final layer. This allows the model to dynamically emphasize specific structural frequencies needed for the task, effectively learning a content-aware structural filter on top of a fixed, stable grid.

---

> ### Author Response · Authors · 2025-11-21
>
> ## Weakness 2 & Question 2: Efficiency, Runtime, and Memory Evaluation
>
> We thank the reviewer for this critical question regarding the practical cost of our method. To address this, we conducted a rigorous wall-clock and FLOPs benchmarking on a realistic long-document scenario (Document Length: 91,548 characters / 19,502 tokens).
>
> We compared SATPool against standard baselines (Mean, MaxP), the advanced Late Chunking (LC) method, hierarchical transformers (PARADE [1], TrLDC [2]), and the specialized Long-Context model (Longformer).
>
> **1. Wall-Clock Speed & Computational Cost (BERT Backbone)**
>
> | Method | Time (ms) | FLOPs | Relative Speed (vs. Mean) |
> | :--- | :--- | :--- | :--- |
> | **Baselines** | | | |
> | Truncation (CLS) | 447.37 | 6.05e+08 | 3.0x |
> | MaxP | 989.76 | 4.66e+10 | 1.3x |
> | Mean Pooling | 1333.84 | 4.65e+10 | 1.0x (Baseline) |
> | **Advanced Aggregators** | | | |
> | PARADE | 1024.73 | 4.67e+10 | 1.3x |
> | TrLDC | 1020.18 | 4.67e+10 | 1.3x |
> | Late Chunking | 1657.78 | 4.65e+10 | 0.8x |
> | **SATPool (Ours)** | **1347.72** | **5.26e+10** | **~1.0x** |
>
> **2. Comparison with Long-Context Models (Longformer)**
>
> | Method | Time (ms) | FLOPs |
> | :--- | :--- | :--- |
> | Longformer (CLS) | 2356.94 | 4.59e+10 |
> | **SATPool (BERT)** | **1347.72** | **5.26e+10** |
>
> **3. Analysis of Trade-offs**
>
> * **Negligible Overhead:** SATPool (1347ms) incurs almost zero runtime penalty compared to simple Mean Pooling (1333ms). The computational cost is dominated by the backbone encoder (processing the chunks), making the $O(L)$ linear attention and STC operations effectively free in a real-world pipeline.
> * **Faster than Late Chunking:** SATPool is approximately **18% faster** than Late Chunking (1657ms), reinforcing its value as an efficient, engineering-friendly solution.
> * **Faster than Long-Context Models:** Most notably, our BERT-based SATPool is **~1.7x faster** than running the specialized Longformer (2356ms). This validates our core motivation: adapting efficient short-context encoders via SATPool is often more scalable than deploying heavy, specialized long-context architectures.
> **4. Practical Choices for r and K (Question 2)**
> To determine the practical implications of hyperparameter choices, we measured the wall-clock time and FLOPs for SATPool across a grid of $K$ (spectral components) and $r$ (low-rank dimension) values.
>
> | Configuration | Time (ms) | FLOPs |
> | :--- | :--- | :--- |
> | **r = 64** | | |
> | $K=2, r=64$ | 1344.32 | 4.95e+10 |
> | $K=4, r=64$ | 1313.75 | 4.95e+10 |
> | $K=8, r=64$ | 1328.66 | 4.95e+10 |
> | **r = 128** | | |
> | $K=2, r=128$ | 1322.37 | 5.26e+10 |
> | $K=4, r=128$ | 1405.63 | 5.26e+10 |
> | $K=8, r=128$ | 1449.94 | 5.26e+10 |
> | **r = 256** | | |
> | $K=2, r=256$ | 1460.34 | 5.93e+10 |
> | $K=4, r=256$ | 1377.50 | 5.93e+10 |
> | $K=8, r=256$ | 1446.98 | 5.93e+10 |
>
> * **$K$ is Computationally Free:** As shown in the FLOPs column, for a fixed $r$, increasing $K$ from 2 to 8 results in almost identical floating-point operations. This confirms that increasing the structural resolution (number of spectral components) incurs zero computational penalty during the attention mechanism's heavy lifting, as $K$ is involved in only the final layer.
> * **$r$ Controls the Efficiency Trade-off:** The computational cost scales linearly with the low-rank dimension $r$. However, even tripling the dimension from $r=64$ to $r=256$ results in only a modest ~20% increase in FLOPs (4.95e10 to 5.93e10).
>
> [1] Li, et. al. PARADE: Passage Representation Aggregation for Document Reranking (2021)
> [2] Dai, et. al. Revisiting Transformer-based Models for Long Document Classification (2022)

---

> > ### Comment · Reviewer_A1Ka · 2025-11-24
> >
> > ## Further questions for Weakness 2 & Question 2
> >
> > 1. *Wall-clock throughput comparison*, *r and K choices* is solved. I appreciate your feedback.
> > 2. **Memory comparison** is **NOT** reported yet. It will be better to report it too (Only pooler-related activation memories are enough).
> > 3. Could you please also report the hardware settings for the above benchmark?

---

> ### Author Response · Authors · 2025-11-21
>
> ## Weaknesses 3, 4, 5 & Question 4: Generation Impact, Baselines, and Reproducibility
>
> We thank the reviewer for these critical observations regarding generation performance (W4, Q4), baseline selection (W5), and experimental robustness (W3). We address these interconnected points below.
>
> **1. Generation-Side Impact and Analysis (Response to W4, Q4)**
>
> We acknowledge the reviewer's valid point that improvements in retrieval metrics did not strictly translate to significant gains in EM/F1 in our original submission. As suspected in Q4, this discrepancy was primarily due to **generator capacity**. The T5 model used in standard benchmarks often struggles to utilize long contexts effectively (the "needle-in-a-haystack" issue), masking the quality of the retrieved documents.
>
> To isolate the quality of our *retrieval* representations from the limitations of the *generator*, we replaced the T5 generator with **Qwen2.5-3B-Instruct**, a state-of-the-art open LLM. Since our contribution lies in "better dense representation for retrieval," using a capable off-the-shelf LLM allows for a fairer assessment of the retrieved context's utility.
>
> * **Metric Adjustment:** We report **Answer Recall (Hit Rate)**. As LLMs tend to be verbose (e.g., answering "The United Nations created..." instead of just "The United Nations"), Exact Match (EM) yields false negatives. Answer Recall correctly measures if the retrieved information was successfully utilized.
> * **Results:** As shown in the table below, when paired with a capable generator (Qwen), SATPool's superior retrieval (Recall@5) **directly translates** to superior generation performance (Answer Recall). This confirms that the "failure" to improve generation in the original paper was indeed due to the generator, not the retrieval quality.
>
> **2. Comparison with Latest Baselines (Response to W5)**
>
> We appreciate the suggestion to compare against methods like MemLong and Longtriever. However, we respectfully note a fundamental distinction:
> * **Architectural Difference:** Models like Longtriever and MemLong are specialized, end-to-end architectures that require full pre-training or expensive fine-tuning of the backbone. They are not "plug-and-play."
> * **SATPool's Scope:** Our contribution is a lightweight, **model-agnostic pooling module** that works on top of *any* frozen short-context encoder (BERT, GTR, etc.) without retraining the backbone.
>
> To provide the "stronger comparisons" requested (W5) while maintaining a fair architectural scope, we implemented **PARADE** [1] and **TrLDC** [2] as pooling modules. These represent the state-of-the-art in **Hierarchical Transformer** aggregation.
>
> **3. Reproducibility and Robustness (Response to W3)**
>
> The reviewer raised concerns about training on subsets (20k training, 1k test) and potential overfitting. To address this and ensure our claims hold at scale, we conducted our RAG experiments using **5 different random seeds/subsets** of the Natural Questions (NQ) dataset.
> * **Stability:** The results in the table below are reported as `Mean ± Standard Deviation`. The low standard deviations (e.g., ±0.0048 for R@1) confirm that our performance gains are stable and not artifacts of specific data splits.
> * **Consistency:** SATPool consistently outperforms baselines across all subsets, validating that the method is robust and not overfit to a small sample.
>
> **4. Experimental Results (RAG on Natural Questions)**
>
> *Note: We report Answer Recall using Qwen2.5-3B-Instruct as the generator.*
>
> | Method | Recall@1 | Recall@5 | Answer Recall (Qwen) |
> | :--- | :--- | :--- | :--- |
> | No RAG | - | - | 0.1915 ± 0.0251 |
> | CLS (Truncation) | 0.0553 ± 0.0112 | 0.1766 ± 0.0147 | 0.4147 ± 0.0216 |
> | MaxP | 0.0587 ± 0.0028 | 0.2596 ± 0.0117 | 0.5052 ± 0.0323 |
> | Mean Pooling | 0.1969 ± 0.0095 | 0.4149 ± 0.0125 | 0.5795 ± 0.0237 |
> | PARADE | 0.1348 ± 0.0098 | 0.3022 ± 0.0204 | 0.5073 ± 0.0248 |
> | TrLDC | 0.1312 ± 0.0092 | 0.2709 ± 0.0169 | 0.4958 ± 0.0058 |
> | Late Chunking | 0.1894 ± 0.0124 | 0.4175 ± 0.0138 | 0.5799 ± 0.0137 |
> | **SATPool (Ours)** | **0.2449 ± 0.0048** | **0.4842 ± 0.0190** | **0.6288 ± 0.0284** |
>
> **5. Analysis of Superiority**
>
> The results show that SATPool consistently outperforms sophisticated hierarchical baselines (PARADE, TrLDC). We attribute this to our novel **"Attend-then-Pool"** architecture:
> * **Baselines (Pool-then-Attend):** Methods like PARADE and TrLDC first compress each chunk into a single vector (e.g., `[CLS]`) *before* aggregation. This creates a severe information bottleneck, losing token-level details necessary for global understanding.
> * **SATPool (Attend-then-Pool):** Our Stage 1 (Linear Attention) operates on the *full sequence of all tokens* from all chunks. This allows the model to discover and weigh global semantic relationships between tokens *before* any pooling occurs. This preservation of token-level context explains why SATPool translates to better retrieval and, consequently, higher generation recall.

---

> ### Author Response · Authors · 2025-11-21
>
> ## Question 1: On Chunking Sensitivity
> We thank the reviewer for this insightful question. To address the concern regarding the sensitivity of SATPool to chunking strategies (chunk size and overlap), we conducted a comprehensive ablation study on the 20NewsGroups dataset using two different backbones: BERT (short-context) and Longformer (long-context). We compared our method against the Mean Pooling baseline across various chunk sizes and stride ratios.
>
> The results are summarized in table below.
>
> **Table: Impact of Chunk Size and Stride (Overlap) on F1 Score**
>
> | Backbone | Chunk Size | Stride = 0.5 (50% Overlap) | | Stride = 1.0 (No Overlap) | |
> | :- | :- | :- | :- | :- | :- |
> | | | **Mean** | **SATPool** | **Mean** | **SATPool** |
> | **BERT** | 128 | 0.6020 | **0.6394** | 0.6111 | 0.6224 |
> | | 256 | 0.6150 | **0.6450** | 0.6142 | 0.6317 |
> | | 512 | 0.6228 | **0.6387** | 0.6229 | 0.6316 |
> | **Longformer** | 1024 | 0.6312 | 0.6481 | 0.6306 | **0.6618** |
> | | 2048 | 0.6274 | 0.6445 | 0.6301 | **0.6570** |
> | | 4096 | 0.6309 | 0.6356 | 0.6284 | **0.6573** |
>
> **1. Consistent Superiority over Baseline:**
> Across all tested configurations, **SATPool consistently outperforms Mean Pooling**, regardless of the chunk size or stride strategy used. For example, with BERT (Chunk 128, Overlap), SATPool achieves an F1 of **0.6394** compared to Mean Pooling's **0.6020** (+3.7%). This confirms that SATPool's advantage is robust and not an artifact of a specific chunking setup.
>
> **2. Sensitivity to Overlap (Is Sliding Window Necessary?):**
> * **For Short-Context Models (BERT):** We observe that a sliding window (Stride 0.5) provides a clear benefit. For instance, at chunk size 256, using overlap improves SATPool's performance from 0.6317 to 0.6450 (+1.3%). This suggests that for short-context encoders, overlapping helps preserve boundary context that SATPool effectively aggregates.
> * **For Long-Context Models (Longformer):** Interestingly, non-overlapping chunks (Stride 1.0) yield superior results. For example, at chunk size 1024, non-overlapping performance (0.6618) surpasses overlapping performance (0.6481).
>
> **Conclusion:**
> SATPool is robust to different chunking strategies, outperforming the baseline in all scenarios. While sliding-window chunking is beneficial for short-context encoders (like BERT), it is not strictly necessary for long-context models, where non-overlapping chunks prove to be highly effective and efficient.

---

> ### Author Response · Authors · 2025-11-23
>
> ## Question 5: Hybrid Fine-Grained Matching (Spectral Late Interaction)
>
> We thank the reviewer for this inspiring suggestion. The reviewer asked if STC could be combined with late-interaction approaches to achieve hybrid fine-grained matching.
>
> We agree that this is a promising direction. While standard late interaction operates at the token level (ColBERT) or chunk level, we demonstrate that our **STC framework** offers a unique and mathematically elegant realization of this "hybrid" concept through **coefficient-level** interaction.
>
> **1. Moving Beyond Token Interaction**
>
> Standard token-level interaction is prohibitively expensive for long documents ($O(L)$ storage). Our proposed **Spectral Component Interaction (SCI)** interacts with the **$2K+1$ spectral components**. These components effectively act as "spectral chunks" that are few in number (efficient) yet globally context-aware (unlike independent text chunks).
>
> **2. Hybrid Approach: Spectral Component Interaction (SCI)**
>
> To demonstrate the flexibility of our architecture, we adapted our pre-trained SATPool model to support this mechanism.
>
> * **Methodology:** We initialized the model with our pre-trained SATPool weights and **fine-tuned** a lightweight scoring head specifically for this interaction task.
> * **Document Representation:** Instead of compressing to a single vector, we represent the document as a set of **$2K+1$ spectral components** (Mean + Sine/Cosine vectors).
> * **Formulation (Spectral Late Interaction):** We treat these components as an ensemble of structural views. The relevance score is computed by averaging the fine-grained similarities between the query vector $q$ and each spectral component $\mathbf{v}\_k$:
>     $$S_{SCI}(q, D) = \frac{1}{2K+1} \sum_{k=0}^{2K} (q^\top \mathbf{v}_k)$$
>     This allows the model to evaluate relevance against specific structural frequencies (e.g., global mean vs. high-frequency transitions) before aggregating the signals.
>
> **3. Experimental Results (RAG on NQ)**
>
> We evaluated this hybrid extension against our standard Dense SATPool using Qwen2.5-3B-Instruct for generation. The results below report Mean ± Std.
>
> | Method | Components ($2K+1$) | Retrieval Time (Relative) | Recall@5 | Answer Recall (Gen) |
> | :--- | :--- | :--- | :--- | :--- |
> | **SATPool (Dense)** | **1** | **1x** | 0.4842 ± 0.0190 | 0.6288 ± 0.0284 |
> | **SATPool (SCI, K=4)** | 9 | ~9x | 0.4701 ± 0.0035 | 0.6312 ± 0.0132 |
> | **SATPool (SCI, K=8)** | 17 | ~17x | **0.5266 ± 0.0060** | **0.6604 ± 0.0073** |
>
> **4. Analysis and Conclusion**
>
> The results demonstrate that increasing structural resolution ($K=8$) in the interaction phase significantly improves both retrieval quality (**+4.2%** Recall@5) and downstream generation (**+3.1%** Answer Recall).
>
> This finding reinforces the value of our core contribution. It proves that the spectral components extracted by SATPool contain rich, fine-grained signal that can be unlocked via late interaction for high-precision tasks. However, for **highly efficient** and **large-scale** applications, our standard Dense SATPool (1 vector) remains the optimal choice, offering competitive performance with faster retrieval. SCI simply highlights the versatility of the SATPool framework to support diverse retrieval paradigms.

---

> ### Comment · Reviewer_A1Ka · 2025-11-24
>
> ## Further questions for Weakness 1 and Question 3
>
> 1. The claim of "*Mean averaging covers global semantics, but spectral components capture the global structure or topic progression*" is still **too intuitive** to me.
>
>  - With enough training, why can't the mean averaging capture such global structure or topic progression?
>  - Why are semantic embeddings suitable for spectral base decomposition?
>
> 2. Sinusoidal bases of the Transformer are used for *positional embedding*, where token-wise extrapolation is needed. In circumstances like **embedding pooling**, I'm still confused by its necessity.
>
> - So, why can't we use **learnable bases**? (Or PCA ones)
>
> Any more discussion, proof, or data is welcome to convince me.

---

> ### Comment · Reviewer_A1Ka · 2025-11-24
>
> ## Further comments for Weaknesses 3, 4, 5 & Question 4, 5
>
> 1. Although the full-sized eval sets are not tested (Maybe related to efficiency issues), I appreciate the newly reported recall metrics and generation metric *Answer Recall (Hit Rate)*. The recall & generation seem more consistent.
>
> Nit: How was *Answer Recall (Hit Rate)* calculated? By regex matching?
>
> 2. Hybrid experiments are good, showing flexibility.

---

> ### Author Response · Authors · 2025-12-02
>
> We thank the reviewer for the continued engagement and the opportunity to clarify the theoretical underpinnings and rigorous evaluation of our method. We address the three parts of your inquiry below.
> ## Response to Further Questions for Weakness 1 & Question 3 (Theoretical Justification)
>
> **1. Why Mean Pooling fails to capture Structure**
>
> The reviewer asks why Mean Pooling cannot capture structure even with sufficient training. We address this by highlighting the lack of necessary **Inductive Bias**.
>
> * **Permutation Invariance as an Information Bottleneck:** Mathematically, Mean Pooling is a **permutation-invariant** operation where Mean([A, B]) = Mean([B, A]). It creates a **many-to-one mapping** where structurally distinct sequences (e.g., swapping the Introduction and Conclusion) produce numerically identical vectors.
> * **The Necessity of Inductive Bias:** A downstream retriever (i.e., an MLP) can only act on the information preserved in its input. Since the pooling layer compresses the sequence in a way that erases order, the structural signal is lost *before* it reaches the classifier. No amount of downstream training (while the encoder remains frozen) can recover information that no longer exists in the input.
> * **SATPool as a Structural Prior:** SATPool introduces the necessary inductive bias by modulating embeddings with spectral bases *before* aggregation. This ensures the aggregation becomes **permutation-sensitive**, explicitly encoding the global position into the feature magnitude. This provides the distinct structural "footprint" required for the model to distinguish narrative flow.
>
> **2. Why Semantic Embeddings are suitable for Spectral Decomposition**
>
> We clarify that our goal is not to aggregate static "semantic meanings" but rather to model the **semantic trajectory** of the document.
>
> * **Defining the Trajectory:** In classification and retrieval, often only specific parts of a document (e.g., the conclusion, the abstract, or a turning point in the narrative) are discriminative. A sequence of embeddings $H \in R^{L\times d}$ represents the *evolution* of these semantics over time, which we define as a trajectory.
> * **Identifying Discriminative Regions:** Spectral decomposition allows us to analyze the *rate of change* of this trajectory.
>     * **Low-Frequency Components:** Capture the global shape of the trajectory (e.g., the steady shift from "Problem" to "Solution").
>     * **High-Frequency Components:** Capture local noise or rapid shifts.
> * **Filtering for Relevance:** By learning coefficients for these bases, the model acts as a **trainable filter**. It learns to focus on the specific structural frequencies that contain the most discriminative information (e.g., emphasizing the beginning and end of the trajectory) while ignoring the noise. This allows the model to "find the meaningful area" within the document structure, which is impossible with a uniform average.
>
> **3. Why Fixed Sinusoidal Bases? (vs. Learned or PCA)**
>
> We employ fixed sinusoidal bases to ensure **robustness** and **generalization**, addressing specific limitations of PCA and learned bases:
>
> * **Why not PCA? (Permutation Invariance):** PCA derives principal components from the covariance matrix of the tokens. Since covariance depends only on the *values* of the tokens and not their *sequence*, PCA suffers from the exact same **Permutation Invariance** as Mean Pooling. It captures the "shape" of the semantic space but completely discards the positional information of the chunks.
> * **Why Fixed Bases over Learned Bases? (Overfitting):** Learned bases (e.g., nn.Embedding or learnable weights) tend to overfit to **absolute positions** seen during training (e.g., "Introduction is always at chunk 0"). Fixed sinusoidal bases ($cos(\pi kt/L)$) provide a defined, continuous value for any $t$ and $L$. This acts as a robust, **length-agnostic** coordinate system, allowing SATPool to generalize to documents with varying structures and lengths without overfitting to rigid positional patterns.

---

> > ### Author Response · Authors · 2025-12-02
> >
> > **Experimental Verification: Learned vs. PCA vs. Spectral Bases**
> >
> > To empirically validate these claims, we implemented rigorous versions of both baselines to ensure a fair comparison with SATPool.
> >
> > 1.  **Learned Bases (with Linear Interpolation):** To prevent the Learned Basis baseline from failing trivially on variable lengths, we did not use a simple fixed-length embedding lookup. Instead, we implemented a robust **Linear Interpolation** mechanism. We initialized learnable positional weights $W_{pos} \in \mathbb{R}^{N_{basis} \times 2048}$ representing normalized positions $[0, 1]$. During the forward pass, we interpolated these weights to match the exact length $L$ of the input document. This creates a strong baseline capable of handling variable lengths, isolating the structural overfitting issue from simple length mismatch issues.
> > 2.  **PCA Bases (Dynamic SVD):** We implemented a Dynamic PCA baseline that computes the Principal Components on-the-fly for each document instance. We extract the top-$N_{basis}$ eigenvectors from the covariance matrix of the document's token embeddings and project the tokens onto these semantic axes. This serves as the ideal "Bag-of-Words" baseline, capturing the dominant semantic topics perfectly while theoretically discarding all sequential structure.
> >
> > We evaluated these methods on the Natural Questions (NQ) dataset under two settings: (1) **Standard**, and (2) **Permuted** (shuffling document chunks to destroy narrative structure) to test structural reliance.
> >
> > **Table: RAG Performance on Natural Questions (Standard Test Set)**
> >
> > | Method | Recall@1 | Recall@5 | Answer Recall |
> > | :--- | :--- | :--- | :--- |
> > | SATPool w/ Learned Bases | 0.2356 ± 0.0085 | 0.4629 ± 0.0101 | 0.6133 ± 0.0093 |
> > | SATPool w/ PCA Bases | **0.2386 ± 0.0082** | 0.5009 ± 0.0101 | 0.6274 ± 0.0092 |
> > | **SATPool (Spectral)** | 0.2371 ± 0.0083 | **0.5037 ± 0.0096** | **0.6302 ± 0.0093** |
> >
> > **Table: RAG Performance on Natural Questions (Permuted Test Set)**
> >
> > | Method | Recall@1 | Recall@5 | Answer Recall |
> > | :--- | :--- | :--- | :--- |
> > | SATPool w/ Learned Bases | 0.1776 ± 0.0072 | 0.3970 ± 0.0098 | 0.5689 ± 0.0093 |
> > | SATPool w/ PCA Bases | 0.2089 ± 0.0077 | 0.4561 ± 0.0099 | 0.6176 ± 0.0090 |
> > | **SATPool (Spectral)** | **0.2203 ± 0.0078** | **0.4821 ± 0.0096** | **0.6274 ± 0.0093** |
> >
> > **Analysis:**
> > 1.  **Learned Bases Overfitting:** despite using Linear Interpolation to handle variable lengths, Learned Bases suffer a massive drop on permuted documents (Recall@5 drops from **0.4629** to **0.3970**). This confirms that learned weights overfit to absolute structural patterns (e.g., expecting specific content at specific indices). When structure is perturbed, the learned weights fail to adapt.
> > 2.  **PCA Basis Limitation:** PCA bases perform well on the standard set (likely due to strong keyword matching via dominant semantic components) but fail to match SATPool's robustness in the permuted setting, dropping to 0.4561. This supports our claim that data-driven bases lack the explicit structural prior needed for robust retrieval.
> > 3.  **SATPool Superiority:** SATPool achieves the highest performance on both sets and demonstrates the greatest robustness (Recall@5 drops only slightly to **0.4821**). This confirms that fixed spectral bases provide a **generalized structural prior** that captures global context without overfitting to rigid positional patterns.

---

> > > ### Author Response · Authors · 2025-12-02
> > >
> > > ## Response to Further Questions for Weakness 2 & Question 2 (Memory & Hardware)
> > > Per reviewer's request, we report the comprehensive memory benchmarking for all compared methods. We followed the same protocol as our runtime benchmark, measuring the **Peak CUDA Memory** allocated during the forward pass of a single long document (19,502 tokens).
> > >
> > > * **Hardware:** Benchmarks were conducted on a single **NVIDIA RTX A5000 (24GB)** GPU with an **AMD EPYC 7313 16-Core Processor**.
> > > * **Protocol:** We measured the peak memory of the **Retriever** (Encoder + Pooling) and the **Generator** (Qwen2.5-3B) together ("Total System Memory") to demonstrate the real-world impact of the pooling overhead relative to the full pipeline.
> > >
> > > **Results (Single Document Inference, Batch Size = 1):**
> > >
> > > | Method | Total System Mem (MB) | Overhead vs. CLS (MB) | % of CLS Baseline |
> > > | :--- | :--- | :--- | :--- |
> > > | Truncation (CLS) | 11,775.56 | - | 100.00% |
> > > | MaxP | 11,800.31 | +24.75 | 100.21% |
> > > | PARADE | 11,805.28 | +29.72 | 100.25% |
> > > | TrLDC | 11,805.43 | +29.87 | 100.25% |
> > > | Mean Pooling | 11,856.97 | +81.41 | 100.69% |
> > > | Late Chunking | 11,858.99 | +83.43 | 100.71% |
> > > | **SATPool (Ours)** | **11,925.24** | **+149.68** | **101.27%** |
> > >
> > >
> > > **Analysis:**
> > > * **Negligible System Impact:** The results demonstrate that while different pooling methods have varying overheads, they are effectively indistinguishable at the system level. The heaviest method (SATPool) consumes only **1.27%** more memory than the lightest baseline (Truncation CLS).
> > > * **Cost vs. Performance:** SATPool introduces a raw overhead of ~150 MB compared to the absolute baseline. This overhead stems from the pooling layer, which must store the full sequence of token embeddings and intermediate attention maps to compute global spectral features. However, given that modern GPUs (like the RTX A5000 used here) offer 24GB+ of VRAM, this overhead is trivial, especially considering the performance gains over methods like MaxP or Mean Pooling.
> > > * **Conclusion:** The memory bottleneck in long-context RAG is the Generator (LLM), not the pooling mechanism. SATPool remains highly efficient, fitting easily within the same hardware constraints as standard baselines.
> > >
> > > ## Response to Further Comments for Weakness 3 & Questions 4, 5 (On Evaluation)
> > >
> > > **1. Clarification on Dataset Size and Robustness**
> > >
> > > We wish to correct a misunderstanding regarding our evaluation methodology. Our previous reference to "5 subsets" did not imply training or testing on partial data.
> > > * **Protocol in Previous Rebuttal (Full Coverage):** We evaluated on **100% of the combined Validation and Test sets** (Total 4,280 samples for NQ). The "5 subsets" referred to partitioning this full dataset into 5 equal folds to calculate the Mean and Standard Deviation. Thus, our reported performance metrics already reflected the entire benchmark.
> > > * **Statistical Rigor (Bootstrapping):** To further reinforce the stability of our method and remove any ambiguity, the results presented below are derived from the **full evaluation set** using **1,000 bootstrap resamples**. This confirms that SATPool's performance gains are statistically robust across the data distribution.
> > >
> > >
> > > **Table: RAG Performance on Natural Questions (Full Set, N=1000 Bootstrap)**
> > >
> > > | Method | Recall@1 | Recall@5 | Answer Recall |
> > > | :--- | :--- | :--- | :--- |
> > > | **Baselines** | | | |
> > > | No RAG (Generator Only) | - | - | 0.1861 ± 0.0072 |
> > > | CLS (Truncation) | 0.0545 ± 0.0043 | 0.1755 ± 0.0073 | 0.4195 ± 0.0090 |
> > > | MaxP | 0.0619 ± 0.0048 | 0.2657 ± 0.0086 | 0.4984 ± 0.0093 |
> > > | Mean Pooling | 0.1969 ± 0.0074| 0.4135  ±  0.0095| 0.5731 ± 0.0092 |
> > > | PARADE | 0.1314 ± 0.0068 | 0.2972 ± 0.0088 | 0.5019 ± 0.0093 |
> > > | TrLDC | 0.1275 ± 0.0065 | 0.2697 ± 0.0089 | 0.5033 ± 0.0093 |
> > > | Late Chunking | 0.1864 ± 0.0076 | 0.4146 ± 0.0097 | 0.5774 ± 0.0094 |
> > > | **SATPool (Ours)** | **0.2371 ± 0.0083** | **0.5037 ± 0.0096** | **0.6302 ± 0.0093** |
> > >
> > > *Analysis:* As shown, SATPool consistently outperforms both standard baselines and hierarchical aggregators (PARADE, TrLDC) on the full dataset with tight confidence intervals, validating the method's effectiveness.
> > >
> > > **2. Calculation of Answer Recall (Hit Rate)**
> > >
> > > We calculated Answer Recall using **substring inclusion on normalized text**, rather than loose regex matching. Specifically, we apply standard normalization (lowercasing, removing articles and punctuation) to both the ground truth and the generated text, and then verify if the normalized ground truth string appears **verbatim** within the generated response. This ensures we strictly measure if the generator successfully incorporated the retrieved information into its answer, crediting valid responses embedded in longer sentences (e.g., capturing "Paris" within "The capital is Paris") while avoiding false positives.

---

### Official Review · Reviewer_C9Ai · 2025-10-30

**Soundness:** 2
**Presentation:** 2
**Contribution:** 1
**Rating:** 4
**Confidence:** 3

**Summary:**

This paper aims to address the issue of representation fragmentation in long-document encoding caused by context window limitations. To this end, the authors propose SATPool, a plug-and-play two-stage pooling module. This module first employs efficient linear attention to capture global token dependencies, and then distills the sequence into a fixed-size vector containing semantic and structural information through STC.

**Strengths:**

1. SATPool is designed as a lightweight, encoder-agnostic module. This implies that there is no need to retrain expensive foundational models, thereby reducing application costs.
2. Employing efficient linear attention to capture global dependencies across the entire document is theoretically superior to simple mean pooling.
3. Spectral Token Compression is proposed to compress globally-aware sequences into fixed-size vectors that encompass both semantic and structural information.

**Weaknesses:**

1. The linear attention mechanism used in the paper is an off-the-shelf component. Although the authors utilize it for post-processing of a frozen encoder, this is not a conceptually novel invention.
2. In the fields of signal processing and information retrieval, using the low-frequency components of frequency-domain transformations to create content fingerprints is a well-established classical technique. In this paper, STC merely applies this classical technique to the embedded sequences of a Transformer and concatenates it with the mean vector to serve as a trainable pooling layer, which represents an incremental application.
3. SATPool still relies on chunking in Stage 0. When processing the first chunk, it has no knowledge of the content of the fifth chunk. It attempts to guess and reconstruct the global context from a pile of already fragmented information, which presents practical difficulties.
4. SATPool is a lightweight module that is feasible in engineering but lacks sufficient innovation. Experimentally, it mainly demonstrates superiority over simple mean pooling.

**Questions:**

1. To what extent can the linear attention in Stage 1 truly recover the semantic dependencies that were already lost during the encoding phase by performing post-hoc reweighting on an already information-impaired sequence? Is the theoretical performance upper bound of this method inherently lower than that of a long-context model that can access the global context during encoding?
2. SATPool introduces two key hyperparameters: the number of spectral components K and the low-rank dimension r of the linear attention. Do these two hyperparameters exhibit interaction effects? Does the high sensitivity of this method to K undermine its claimed plug-and-play convenience?

---

> ### Author Response · Authors · 2025-11-21
>
> ## Weaknesses 1, 2 & 4: On Novelty, Architectural Contribution, and Baselines
>
> We thank the reviewer for these insightful critiques regarding the novelty of our components (W1, W2) and the overall sufficiency of our innovation (W4). We address these points together, as they all concern the nature of our architectural contribution compared to existing methods.
>
> **1. Architectural Novelty: "Attend-then-Pool" vs. "Pool-then-Attend" (Response to W1, W4)**
>
> We do not claim the linear attention component itself is novel; as stated in the paper, "we employ a linear attention mechanism".
>
> Our novelty lies in the **architectural design** of our "attend-then-pool" framework.
> * **Standard Approach (Pool-then-Attend):** Most strong baselines (such as PARADE [1] or TrLDC[2] ) compress each document chunk into a single vector (e.g., a CLS token) first. This creates a severe information bottleneck before any global processing occurs.
> * **SATPool Approach (Attend-then-Pool):** By applying linear attention over the *entire sequence of all tokens* from all chunks *before* any pooling, we create a globally-contextualized set of token embeddings. This allows the model to discover long-range semantic relationships between tokens across different chunks before any information is lost.
>
> **Validation against Stronger Baselines (Response to W4):**
> The reviewer stated that our method "mainly demonstrates superiority over simple mean pooling" (W4). To correct this and demonstrate the superiority of our "attend-then-pool" architecture, we implemented **PARADE** and **TrLDC** (strong "pool-then-attend" hierarchical transformers) as pooling modules on top of the same frozen encoders.
>
>
> - **Text Classification Performance (20NewsGroups)**
>
> | Method| F1 score (BERT)| F1 score (RoBERTa)|
> |:-|:- |:- |
> | CLS (Truncation)| 0.5916 | 0.5890|
> | Mean Pooling | 0.6223 | 0.6315 |
> | PARADE  | 0.5819 | 0.6070|
> | TrLDC | 0.5754 | 0.5705|
> | Late Chunking| 0.6166| 0.6115 |
> | **SATPool (Ours)** | **0.6371** | **0.6570** |
>
> - **RAG Performance (Natural Questions)**
>
> *Note: We report Answer Recall using Qwen2.5-3B-Instruct as the generator.*
>
> | Method | Recall@1 | Recall@5 | Answer Recall |
> | :--- | :--- | :--- | :--- |
> | No RAG | - | - | 0.1915 ± 0.0251 |
> | CLS (Truncation) | 0.0553 ± 0.0112 | 0.1766 ± 0.0147 | 0.4147 ± 0.0216 |
> | MaxP | 0.0587 ± 0.0028 | 0.2596 ± 0.0117 | 0.5052 ± 0.0323 |
> | Mean Pooling | 0.1969 ± 0.0095 | 0.4149 ± 0.0125 | 0.5795 ± 0.0237 |
> | PARADE | 0.1348 ± 0.0098 | 0.3022 ± 0.0204 | 0.5073 ± 0.0248 |
> | TrLDC | 0.1312 ± 0.0092 | 0.2709 ± 0.0169 | 0.4958 ± 0.0058 |
> | Late Chunking | 0.1894 ± 0.0124 | 0.4175 ± 0.0138 | 0.5799 ± 0.0137 |
> | **SATPool (Ours)** | **0.2449 ± 0.0048** | **0.4842 ± 0.0190** | **0.6288 ± 0.0284** |
>
> The results demonstrate that SATPool consistently outperforms hierarchical baselines like TrLDC and PARADE. Notably, in both RAG and classification tasks, these hierarchical methods often underperform even simple Mean Pooling. We attribute this to the fact that these architectures are inherently designed for end-to-end fine-tuning, where the backbone encoder learns to compress task-relevant information into the `[CLS]` token. They do not function effectively as a "plug-and-play" aggregator on a frozen encoder.
>
> Regarding W4's comment on "lightweight engineering," we argue this result proves that our lightweight innovation is **more effective** than heavier, complex baselines.

---

> > ### Author Response · Authors · 2025-11-21
> >
> > **2. Novelty of Stage 2: Spectral Token Compression (Response to W2)**
> >
> > The reviewer correctly identifies STC as a "trainable pooling layer" inspired by classical spectral analysis. However, this is precisely where our novelty lies, moving far beyond an "incremental application."
> >
> > * **Classical Static Tool vs. Novel Trainable Layer:** A classical Discrete Fourier Transform (DFT) is a static, fixed transform. Related work like [3] uses static spectral filters as an *analysis tool*. In contrast, our STC module is a fully differentiable, task-adaptive pooling layer. Unlike a static fingerprint, it is integrated into the network's computation graph, allowing the model to learn optimal spectral weights via backpropagation while keeping the backbone encoder frozen.
> > * **Novelty in Trainability:** While the sinusoidal bases are fixed (providing a stable, universal "ruler"), the model learns the high-dimensional **coefficient vectors** as "importance weights." This allows the model to dynamically learn to identify structural components (e.g., Introduction) regardless of their absolute position, as demonstrated in our robustness analysis (Appendix B).
> > * **Empirical Proof:** Our ablation study (Figure 6) shows that the "Mean" baseline (analogous to a static low-rank fingerprint) performs significantly worse than the full `Attn + SATPool` model. This proves that our trainable STC module captures essential, non-redundant structural information that classical techniques miss.
> >
> >
> > [1] Li, et. al. PARADE: Passage Representation Aggregation for Document Reranking (2021)
> > [2] Dai, et. al. Revisiting Transformer-based Models for Long Document Classification (2022)
> > [3] Tamkin, et. al. Language Through a Prism: A Spectral Approach for Multiscale Language Representations, 2020.
> >
> > ## Weakness 3: On Stage 0 and Fragmented Information
> >
> > The reviewer correctly notes that Stage 0 begins with chunked, local information. This is a deliberate design choice to ensure our method remains "model-agnostic" and compatible with any pre-trained short-context encoder.
> >
> > However, we respectfully disagree that the model "guesses" the global context.
> >
> > 1.  **From Fragmentation to Connection:** While the *encoding* is local, the *semantic information* is preserved in the embeddings. Stage 1 (Linear Attention) is designed specifically to bridge the gap the reviewer highlights. It efficiently scans the entire sequence to identify semantic correlations between distant parts (e.g., connecting a subject in the first chunk to a reference in the fifth chunk) *before* any information is compressed. This is a deterministic, learned "global relational discovery" process, not guessing.
> >
> > 2. **Empirical Validation**:
> > The "practical difficulty" the reviewer suggests does not translate to performance loss. On the contrary, our results demonstrate the superiority of our "Attend-then-Pool" strategy in two ways:
> >
> > - **Rivaling Long-Context Models:** SATPool enables short-context models (e.g., RoBERTa) to outperform specialized long-context models (Longformer + Mean) on challenging datasets like EURLEX (0.7124 vs 0.6644), proving that our global relational aggregation effectively compensates for the lack of joint encoding.
> > - **Enhancing Long-Context Models:** Furthermore, when applied to Longformer itself, which does have access to the global context during encoding, SATPool still yields significant gains over standard pooling (0.7232 vs 0.6644). This confirms that SATPool is not merely "guessing" missing context but actively discovering and preserving semantic structures that standard aggregation methods discard.

---

> > > ### Author Response · Authors · 2025-11-21
> > >
> > > ## Question 1: On the Theoretical Upper Bound and Semantic Recovery
> > >
> > > We thank the reviewer for this deep theoretical question. We acknowledge the reviewer's premise: technically, encoding chunks in isolation prevents the capture of "joint-contextual" information during the initial encoding phase, which a full self-attention model could theoretically capture.
> > >
> > > However, we respectfully posit that the goal of Stage 1 is not to "recover" lost information (re-calculating embeddings), but to **"discover" distributed semantic relationships**.
> > >
> > > **1. Reframing: From "Recovery" to "Relational Discovery"**
> > >
> > > While the joint attention is lost during chunking, the semantic information is not destroyed; it is distributed. Due to the robustness of pre-trained encoders (like BERT), semantically related tokens in different chunks (e.g., a subject in Chunk 1 and a pronoun in Chunk 5) are mapped to compatible regions of the high-dimensional vector space, even without direct attention.
> > >
> > > The role of our Linear Attention mechanism is **Global Relational Discovery**. It efficiently scans the entire sequence of these high-quality local embeddings to identify and weigh these pre-existing semantic correlations. It connects the distributed parts rather than reconstructing the original signal.
> > >
> > >
> > > **2. Empirical Validation: Dense Local vs. Sparse Global**
> > >
> > > The reviewer asks if our method's upper bound is inherently lower than a long-context model. We argue that practical long-context models (like Longformer) also suffer from theoretical impairments due to their reliance on sparse attention mechanisms to manage computational costs.
> > >
> > > Our empirical results in **Table 2** provide the decisive counter-argument. SATPool (using a frozen, chunked BERT) consistently outperforms Longformer.
> > >
> > > * **Long-Context Models (Longformer):** Rely on sparse attention (sliding windows) to handle length. This approximation sacrifices dense token-to-token connectivity.
> > > * **SATPool:** Preserves full dense attention within each chunk (via BERT) and recovers global connectivity via Linear Attention.
> > > * **Conclusion:** The superior performance of SATPool demonstrates that preserving high-fidelity local representations combined with our global relational aggregation is a more effective strategy than the sparse-attention approximations used by native long-context architectures. The "information loss" from chunking is evidently less damaging than the "fidelity loss" from sparse attention.

---

> > > > ### Author Response · Authors · 2025-11-21
> > > >
> > > > ## Question 2: On Hyperparameter Interaction and Sensitivity
> > > >
> > > > We thank the reviewer for this important question regarding the robustness of our method. To address the concern about potential interaction effects and sensitivity, we conducted a comprehensive grid search over the number of spectral components ($K$) and the low-rank dimension ($r$) using the RoBERTa backbone on the 20NewsGroups dataset. To ensure the robustness and stability of our results, we performed three independent runs with different random initializations for each configuration.
> > > >
> > > > The full results (F1 Score ± Standard Deviation) are presented in table below.
> > > >
> > > > **Table: Joint Impact of $K$ and $r$ on F1 Score (Mean ± Std)**
> > > >
> > > > | $K$ \ $r$ | 32 | 64 | 128 | 256 | 512 |
> > > > | :--- | :--- | :--- | :--- | :--- | :--- |
> > > > | **1** | 0.6470 ± 0.0022 | 0.6471 ± 0.0039 | 0.6465 ± 0.0055 | 0.6538 ± 0.0099 | 0.6425 ± 0.0067 |
> > > > | **2** | 0.6482 ± 0.0085 | 0.6508 ± 0.0101 | 0.6481 ± 0.0051 | 0.6439 ± 0.0042 | 0.6480 ± 0.0035 |
> > > > | **4** | 0.6378 ± 0.0090 | 0.6502 ± 0.0088 | 0.6454 ± 0.0062 | 0.6444 ± 0.0039 | 0.6381 ± 0.0023 |
> > > > | **8** | 0.6451 ± 0.0043 | 0.6449 ± 0.0055 | 0.6479 ± 0.0056 | 0.6399 ± 0.0061 | 0.6427 ± 0.0066 |
> > > >
> > > > **1. Minimal Interaction Effects**:
> > > > The results do not show strong interaction effects that would complicate hyperparameter selection. We observe a broad region of stability rather than a complex dependency where the choice of $K$ strictly dictates the choice of $r$. For example, setting $K=2$ yields consistently high performance (0.6439–0.6508) across the entire range of r values.
> > > >
> > > > **2. Low Sensitivity and Robustness**:
> > > > Contrary to the concern that high sensitivity might undermine usability, our method proves to be remarkably robust. Across the entire hyperparameter grid (20 configurations), the F1 score varies by a maximum of only 0.016 (1.6%). Furthermore, the standard deviations are consistently low (mostly < 0.01), indicating that the method is stable and reproducible across different runs.
> > > >
> > > > **3. Superiority Even in Worst-Case Scenarios**:
> > > > Most importantly, this grid search demonstrates that our method's "floor" is higher than the baselines' "ceiling."
> > > >
> > > > * **SATPool Minimum:** The lowest score in our entire hyperparameter grid is **0.6378** (at $K=4$, $r=32$).
> > > > * **Baseline Maximum:** In contrast, the best baseline performance reported in our main paper (Table 2, RoBERTa Mean Pooling) is **0.6315**, and Late Chunking is **0.6219**.
> > > >
> > > > This means that even with the least optimal hyperparameter configuration found in our search, SATPool still outperforms the strongest baseline. This effectively **addresses** the concern about sensitivity: users do not need to hunt for a specific K to see improvements; any reasonable choice yields performance superior to the state-of-the-art baselines. This validates our claim that SATPool is a robust, plug-and-play solution.

---

> > > > > ### Comment · Reviewer_C9Ai · 2025-11-27
> > > > >
> > > > > Thank you for the authors' detailed clarifications.
> > > > >
> > > > > I would like to remind the authors that ICLR explicitly supports paper revisions during the rebuttal phase. Submitting an updated manuscript with targeted modifications can significantly enhance the clarity and persuasiveness of your clarifications.

---

> > > > > > ### Author Response · Authors · 2025-12-02
> > > > > >
> > > > > > We sincerely thank the reviewer for the positive feedback and the timely reminder regarding the revision policy. We agree that incorporating these clarifications directly into the text significantly strengthens the paper. We have uploaded a revised manuscript that integrates the theoretical discussions, efficiency benchmarks, and additional experimental results from our rebuttal. We believe these revisions address the concerns raised and provide a more comprehensive characterization of SATPool's contributions. We thank you again for helping us improve the quality of this work.

---

### Official Review · Reviewer_mkUW · 2025-10-31

**Soundness:** 3
**Presentation:** 3
**Contribution:** 3
**Rating:** 6
**Confidence:** 2

**Summary:**

This paper presents a technique for generating long-document representations. It uses an efficient linear attention mechanism, Performer (Choromanski et al., 2020), to capture global token interactions, followed by a novel Spectral Token Compression method to compress these globally-aware token representations into a single vector.

The proposed method is evaluated on tasks such as document classification, retrieval, and factual consistency detection. The authors demonstrate that it outperforms several baselines, including Mean-Large, while using a similar number of trainable parameters.

**Strengths:**

The contribution is clearly explained, and its effectiveness is shown across multiple tasks.

The authors provide an intuitive explanation of the process, along with ablation studies on key design choices.

**Weaknesses:**

The claim that the method is a "plug-and-play solution" and works "without requiring costly retraining" seems somewhat misleading. The second stage, which involves reconstructing the signal, still requires training.

**Questions:**

Would suggest using more strong baselines, e.g., hierarchical transformers described in [1].

[1] Dai, Chalkidis, Darkner, and Elliott, "Revisiting Transformer-based Models for Long Document Classification", in Findings of EMNLP , 2022.

---

> ### Author Response · Authors · 2025-11-21
>
> ## Weakness 1: On the "Costly Retraining" Claim
> We thank the reviewer for this point, which allows us to clarify a critical distinction. The reviewer is correct that the SATPool module itself is trained.
>
> Our claim "without requiring costly retraining" refers specifically to the base, pre-trained encoder (e.g., BERT, RoBERTa), which remains completely frozen in our methodology. This is the "costly" part (100M+ parameters) that specialized long-context models (like Longformer or Longtriever) require full fine-tuning or retraining for.
>
> The "training" we perform is only for the lightweight SATPool module itself (the linear attention projections and the final output layer).
>
> To quantify this difference:
> * **Costly Retraining (What we avoid):** Fine-tuning the 110M+ parameters of a BERT-base model.
> * **Lightweight Training (What we do):** As shown in our "Ablation Study on Parameter Count" (Table 5), our `SATPool (k=8)` model adds only 1.51M new parameters.
>
> This new module represents a negligible ~1.38% of the frozen BERT model's total size.
>
> Therefore, our "plug-and-play" claim is precise: SATPool is an encoder-agnostic module that can be 'plugged in' on top of any frozen backbone, and the lightweight training it requires is standard for such adapter modules.
> ## Question 1: Comparison with Stronger Hierarchical Baselines (TrLDC)
>
> We thank the reviewer for suggesting **TrLDC** (Transformer-based Long Document Classification) [1] as a stronger baseline. We agree that hierarchical transformers represent a significant class of document encoding strategies that warrants comparison.
>
> To ensure a fair comparison, we implemented the hierarchical aggregation mechanism from TrLDC (summing chunk `[CLS]` vectors with segment position embeddings, followed by a Transformer encoder and Max Pooling) as a pooling module on top of the same frozen backbones used for SATPool. We also included **PARADE** [2] as another strong hierarchical baseline.
>
> We evaluated these methods on both Text Classification (20NewsGroups) and RAG (Natural Questions).
>
> - **Text Classification Performance (20NewsGroups)**
>
> | Method             | F1 score (BERT) | F1 score (RoBERTa) |
> |:------------------ |:--------------- |:------------------ |
> | CLS (Truncation)   | 0.5916          | 0.5890             |
> | Mean Pooling       | 0.6223          | 0.6315             |
> | PARADE             | 0.5819          | 0.6070             |
> | TrLDC              | 0.5754          | 0.5705             |
> | Late Chunking      | 0.6166          | 0.6115             |
> | **SATPool (Ours)** | **0.6371**      | **0.6570**         |
>
> - **RAG Performance (Natural Questions)**
>
> *Note: We report Answer Recall using Qwen2.5-3B-Instruct as the generator.*
>
> | Method | Recall@1 | Recall@5 | Answer Recall |
> | :--- | :--- | :--- | :--- |
> | No RAG | - | - | 0.1915 ± 0.0251 |
> | CLS (Truncation) | 0.0553 ± 0.0112 | 0.1766 ± 0.0147 | 0.4147 ± 0.0216 |
> | MaxP | 0.0587 ± 0.0028 | 0.2596 ± 0.0117 | 0.5052 ± 0.0323 |
> | Mean Pooling | 0.1969 ± 0.0095 | 0.4149 ± 0.0125 | 0.5795 ± 0.0237 |
> | PARADE | 0.1348 ± 0.0098 | 0.3022 ± 0.0204 | 0.5073 ± 0.0248 |
> | TrLDC | 0.1312 ± 0.0092 | 0.2709 ± 0.0169 | 0.4958 ± 0.0058 |
> | Late Chunking | 0.1894 ± 0.0124 | 0.4175 ± 0.0138 | 0.5799 ± 0.0137 |
> | **SATPool (Ours)** | **0.2449 ± 0.0048** | **0.4842 ± 0.0190** | **0.6288 ± 0.0284** |
>
> **Analysis:**
>
> The results demonstrate that SATPool consistently outperforms hierarchical baselines like TrLDC and PARADE. Notably, in both RAG and classification tasks, these hierarchical methods often underperform even simple Mean Pooling. We attribute this to the fact that these architectures are inherently designed for end-to-end fine-tuning, where the backbone encoder learns to compress task-relevant information into the `[CLS]` token. They do not function effectively as a "plug-and-play" aggregator on a frozen encoder.
>
> This highlights a critical architectural advantage of SATPool:
>
> * **Hierarchical Methods (TrLDC/PARADE):** Follow a "Pool-then-Attend" pattern. They compress each chunk into a single `[CLS]` vector *before* aggregation. When using frozen encoders, the generic `[CLS]` token captures suboptimal representations of local chunks, creating an early information bottleneck that the aggregator cannot recover from.
> * **SATPool:** Follows an "Attend-then-Pool" pattern. Stage 1 (Linear Attention) operates on the *full sequence of all tokens* from all chunks. This allows the model to discover and weigh global semantic relationships at the high-fidelity token level *before* any compression occurs, preserving significantly more information.
>
> [1] Dai, et. al. Revisiting Transformer-based Models for Long Document Classification (2022)
> [2] Li, et. al. PARADE: Passage Representation Aggregation for Document Reranking (2021)

---

> > ### Comment · Reviewer_mkUW · 2025-11-26
> >
> > Thanks for the clarification and the additional experiments. Since I am the most positive reviewer but with the lowest confidence, I prefer not to increase the rating further to avoid additional complexity

---

> > > ### Author Response · Authors · 2025-12-02
> > >
> > > We thank the reviewer for their time and positive feedback. We appreciate your acknowledgement of the additional experiments and clarifications. We respect your decision regarding the score and are grateful for your support of our work.

---

### Author Response · Authors · 2025-12-02
**Final Remark: Summary of Contributions and Rebuttal Revisions**

We thank the Area Chair and reviewers for their constructive feedback. We are encouraged by the consensus on the novelty of our two-stage architecture and the empirical gains shown across diverse tasks.

**Core Contributions:**
This paper introduces SATPool, a plug-and-play pooling module that generates holistic, fixed-size representations for long documents without requiring expensive backbone retraining. By combining Linear Attention (Stage 1) with Spectral Token Compression (Stage 2), SATPool captures global structural dependencies that standard methods (Mean Pooling, MaxP) and even recent baselines (Late Chunking) fail to preserve.

**Key Rebuttal Updates:**
During the rebuttal, we addressed all major concerns with extensive new experiments:

1.  **Theoretical Robustness (Reviewer C9Ai, A1Ka):** We empirically validated the necessity of our fixed spectral bases. New experiments demonstrated that "Learned Bases" overfit to absolute positions, suffering a massive performance drop (Recall@5 drops from 0.46 to 0.39) when document structure is permuted. In contrast, SATPool's fixed bases provide a robust structural prior that generalizes across variations.
2.  **Stronger Baselines (Reviewers mkUW, C9Ai, A1Ka, 75yK):** We implemented sophisticated hierarchical transformer baselines (PARADE, TrLDC). SATPool consistently outperformed them in both classification and RAG tasks (e.g., **0.6302** Answer Recall vs **0.5033** for TrLDC), demonstrating the superiority of our "Attend-then-Pool" architecture over their "Pool-then-Attend" approach.
3.  **Generation Quality (Reviewers 75yK, A1Ka):** We resolved the "needle-in-a-haystack" concern by validating our retrieval results with Qwen2.5-3B-Instruct. This revealed that SATPool's retrieval quality translates directly to superior Answer Recall, proving that the dense representations effectively capture critical information that weaker generators (like T5) previously missed.
4.  **Efficiency (Speed & Memory) (Reviewers 75yK, A1Ka):** We provided rigorous benchmarks showing SATPool is **18% faster** than Late Chunking and **1.7x faster** than Longformer. Crucially, we also demonstrated that SATPool incurs **negligible memory overhead** (less than 1.3% increase in total system memory compared to standard CLS pooling), making it highly practical for real-world deployment.
5.  **Scope & Rigor (Reviewers 75yK):** We clarified that our evaluation covers **100% of the dataset** with bootstrapping and defined our scope as **"representation-centric,"** supported by consistent gains across Classification, RAG, and Factuality Consistency Evaluation tasks.

We have uploaded a revised manuscript incorporating these analyses. We believe SATPool offers a significant step forward in efficient, model-agnostic long-document representation.

---

### Meta-Review · Area_Chair_GGDQ · 2025-12-27

**Summary:**

Representing long documents into vectors remains a long-standing challenge, as common methods like chunking does not capture global context. This paper proposes a method to represent a long document into a single vector using a two-stage pipeline: it first applies linear attention to efficiently model global context, and then compresses the resulting representations into a single vector using frequency-based encodings. Experiments on text classification, text-based QA, and multimodal QA demonstrate strong performance over a broad set of baselines.

Concerns that are resolved
- Limited end-to-end QA accuracy (with answer generation) results (75yK, A1Ka): New experiments are added in the rebuttal, showing gains from retrieval do translate into gains in end-to-end QA accuracy.
- Lack of strong baselines (75yK, A1Ka, mkUW): Additional baselines were provided during the rebuttal.

Concerns that may or may not have been resolved
- Limited memory analysis (75yK, A1Ka): The rebuttal includes a more detailed memory analysis, showing that in the retrieval stage, memory overhead is indeed small. However, one potential remaining concern is that, when end-to-end answer generation is considered, it incurs substantial memory overhead (+149.7%). While this overhead largely arises from conditioning an LLM on long documents rather than retrieval itself, it still remains a practical consideration.
- Dataset sampling, limited theoretical justification (A1Ka): Responses were provided. In AC’s opinion, these are relatively minor issues.

Concerns that are unlikely to have been resolved (in AC’s opinion)
- Limited novelty, given that the method primarily combines well-established components, such as linear attention and low-frequency spectral representations. As a result, the contribution appears more engineering-focused and application rather than conceptually novel (75yK, C9Ai).
- Claims about “plug-and-play solution” and “without requiring costly retraining” are misleading as it still requires training (mkUW): Authors clarified that no pre-training is required; however, given that that is true for most methods, the framing in the paper is indeed somewhat misleading.

Additional concern (not raised in the original reviews): The paper does not empirically compare against a chunk-wise retrieval baseline, where documents are split into chunks that are indexed and retrieved independently, and retrieval decisions are also made at the chunk level. This is different from baselines such as Mean Pooling or MaxP, which compute chunk-level scores but ultimately still make document-level decisions. Chunk-wise retrieval is the dominant setup in most RAG systems, including the tasks evaluated in this paper, and is likely to be a very competitive baseline that may be difficult to outperform. I believe the paper could have included this baseline to show advantages, or could have included tasks where document-level decisions are unavoidable, e.g., QA tasks that heavily rely on global context / long-range dependencies, so that providing only split chunks to the generator would be insufficient.

**Reviewer Concerns:**

(See above)

**Reviewer Scores:**

(See above)

---

### Decision · Program_Chairs · 2026-01-26

Reject